# CONFORMAL PREDICTION VIA REGRESSION-AS-CLASSIFICATION

**Etash Guha**
RIKEN Center for AI Project, SambaNova Systems
{etash.guha}@sambanovasystems.com

**Shlok Natarajan**
Salesforce
{shloknatarajan}@salesforce.com

**Thomas Möllenhoff**
RIKEN Center for AI Project
{thomas.moellenhoff}@riken.jp

**Mohammad Emtiyaz Khan**
RIKEN Center for AI Project
{emtiyaz.khan}@riken.jp

**Eugene Ndiaye**
Apple
{e_ndiaye}@apple.com

## ABSTRACT

Conformal prediction (CP) for regression can be challenging, especially when the output distribution is heteroscedastic, multimodal, or skewed. Some of the issues can be addressed by estimating a distribution over the output, but in reality, such approaches can be sensitive to estimation error and yield unstable intervals. Here, we circumvent the challenges by converting regression to a classification problem and then use CP for classification to obtain CP sets for regression. To preserve the ordering of the continuous-output space, we design a new loss function and make necessary modifications to the CP classification techniques. Empirical results on many benchmarks shows that this simple approach gives surprisingly good results on many practical problems.

## 1 INTRODUCTION

Quantifying and estimating the uncertainty of machine-learning models is an important task for many problems, especially mission-critical applications where reliable predictions are required. Conformal Prediction (CP) (Vovk et al., 2005) has recently gained popularity and has been used successfully in applications such as breast cancer detection (Lambrou et al., 2009), stroke risk prediction (Lambrou et al., 2010), and drug discovery (Cortés-Ciriano & Bender, 2020). Under mild conditions, CP techniques aim to construct a prediction set that, for given test inputs, is guaranteed to contain the true (unknown) output with high probability. The set is built using a *conformity score*, which, roughly speaking, indicates the similarity between a new test example and the training examples. The conformal set merely gathers examples that have large conformity scores. Despite its popularity, CP for regression can be challenging, especially when the output distribution is heteroscedastic, multimodal, or skewed (Lei & Wasserman, 2014). The main challenge lies in the design of the *conformity score*. It is common to use a simple choice for score functions such as distance to mean regressor, but such choices may ignore the subtle features of the shape of the output distribution. For instance, this could lead to symmetric intervals or ignoring the heteroscedasticity. In theory, it is better to estimate the (conditional) distribution over the output, for example, by using kernel density estimation and directly using it to build a confidence interval. However, such estimation approaches are also challenging, and estimates can be sensitive to the choice of kernel and hyperparameters, which can yield unstable results.

We circumvent the challenges by exploiting the existing CP techniques for classification. We proceed by first converting regression to a classification problem and then using CP techniques for classification to obtain a conformal set. Regression-as-classification approaches are popular for various applications in computer vision and have led to more accurate training than only-regression training (Stewart et al., 2023; Zhang et al., 2016; Fu et al., 2018; Rothe et al., 2015; Van Den Oord

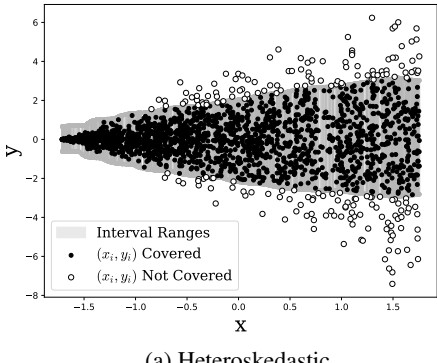
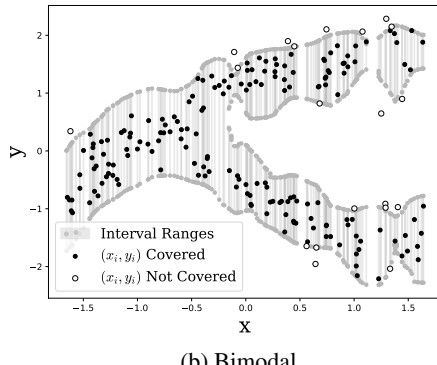

(a) Heteroskedastic                                    (b) Bimodal

Figure 1: We show two examples where the output distribution is heteroskedastic (left) and bimodal (right). In both cases, our method is able to change the interval (shaded gray region) adaptively as the input values $x$ are increased. Examples outside the gray regions (white dots) are deemed different from those inside it (black dots).

et al., 2016; Diaz & Marathe, 2019). We leverage them to construct a distribution-based conformal set that can flexibly capture the shape of the output distribution while preserving the simplicity and efficiency of CP for classification. First, we discretize the output space into bins, treating each bin as a distinct class. Second, to preserve the ordering of the continuous output space, we design an alternative loss function that penalizes the density on bins far from the true output value but also facilitates variability by using an entropy regularization. The loss design is similar in spirit to Weigend & Srivastava (1995); Diaz & Marathe (2019). The resulting method can adapt to heteroscedasticity, bimodality, or both in the label distribution. We verify this on synthetic and real datasets where we achieve the shortest intervals compared to other CP baselines. See examples in Figure 1.

## 2 BACKGROUND ON CONFORMAL PREDICTION

Given a new input $x_{\text{new}}$, CP techniques aim to construct a set that contains the true but unknown output $y_{\text{new}}$ with high probability. Assuming that a pair of input-output variables $(x, y)$ has a joint density $p(x, y)$ and a conditional density $p(y \mid x)$, oracle prediction sets (with joint and conditional coverage) for the output $y$ can be constructed as

$$\{z \in \mathbb{R} : p(x, z) \geq \tau_\alpha\} \text{ or } \{z \in \mathbb{R} : p(z \mid x) \geq \tau_{\alpha, x}\}, \tag{1}$$

where the thresholds $\tau_\alpha$ and $\tau_{\alpha, x}$ are selected to ensure that the corresponding sets have a probability mass that meets or exceeds prescribed confidence level $1 - \alpha \in (0, 1)$. As the ground-truth distribution is unknown, we rely solely on estimating these uncertainty sets using the density estimators $\hat{p}(x, y)$ and $\hat{p}(y \mid x)$. The latter can be inaccurate due to numerous sources of errors such as model misspecification, small sample size, high optimization errors during training, and overfitting. Without a stronger distribution assumption, the finite-sample guarantee is typically not upheld.

Conformal Prediction has arisen as a method for yielding sets that do hold finite-sample guarantees. Given a partially observed instance $(x_{\text{new}}, y_{\text{new}})$ where $y_{\text{new}}$ is unknown, Conformal Prediction (CP) (Vovk et al., 2005) constructs a set of values that contains $y_{\text{new}}$ with high probability without knowing the underlying data distribution. Under conformal prediction, this property is guaranteed under the mild assumption that the data satisfies exchangeability. The set is called the conformal set and is built using a conformity score, denoted by $\sigma(x, y)$, which measures how appropriate an output value is for a given input example. There are many ways to build the conformity score, but they all involve splitting the data into a training set $\mathcal{D}_{\text{tr}}$ and a calibration set $\mathcal{D}_{\text{cal}}$. Often, a prediction model $\mu_{\text{tr}}(x)$ is built using the training set, and then a conformity score is obtained using this model along with the calibration set (we will shortly give an example). The conformal set merely gathers the points with larger conformity scores:

$$\{z \in \mathbb{R} : \sigma(x_{\text{new}}, z) \geq Q_{1-\alpha}(\mathcal{D}_{\text{cal}})\},$$

where $Q_{1-\alpha}(\mathcal{D}_{\text{cal}})$ is the $(1-\alpha)$-quantile of the conformity scores on the calibration data. This set provably contains $y_{\text{new}}$ with probability larger than $1-\alpha$ for any finite sample size and without assumption on the ground-truth distribution.

There are many design choices for this conformity score. For example, one can choose a prediction model $\mu_{\text{tr}}(x)$ as an estimate of the conditional expectation and measure conformity as the absolute value of the residual, i.e., $\sigma(x,y) = -|y - \mu_{\text{tr}}(x)|$. The corresponding conformal set is a single interval centered around the prediction $\mu_{\text{tr}}(x)$ and of constant length $Q_{1-\alpha}(\mathcal{D}_{\text{cal}})$ for any example $x_{\text{new}}$, without taking into account its variability. However, in situations where the underlying data distribution demonstrates skewness or heteroscedasticity, we may desire a more flexible conformity score. We wish to generate an approach to approximate this density function that is versatile in the distributions it can represent and avoids the difficulties seen in prior methods. We explore established density estimation in Classification Conformal Prediction that are already performing effectively.

**Related Works**  Since their introduction, a lot of work has been done to improve the set of conformal predictions. As simple score function, distance to conditional mean ie $\sigma(x,y) = |y - \mu(x)|$ where $\mu(x)$ is an estimate of $\mathbb{E}(y \mid x)$ was prominently used (Papadopoulos et al., 2002; Lei et al., 2018; Guha et al., 2023). Instead, (Romano et al., 2019) suggests estimating a conditional quantile instead and a conformity score function based on the distance from a trained quantile regressor, i.e. $\sigma(x,y) = \max(\mu_{\alpha/2}(x) - y, y - \mu_{1-\alpha/2}(x))$ where $\mu_\alpha$ are the $\alpha$-th quantile regressors. Within the literature, our strategy is more closely related to the distribution-based methods (Lei et al., 2013). Following a similar line of work (Chernozhukov et al., 2021) argued that the conformal quantile regression score function might be less adaptive since the distance of the quantile behaves similarly to the distance of the mean estimate. Instead, they suggested estimating the cumulative (conditional) distribution function and directly outputting $\{y : F(x,y) \in [\alpha/2, 1-\alpha/2]\}$. However, their method cannot account for bimodality since it can only output a single interval by design. An equally interesting approach for distribution-based prediction sets is based on learning a Bayesian estimator, which, however, may require a well-specified prior and can be computationally expensive (Fong & Holmes, 2021). Variants based on estimating the conditional density of the response using histogram regressors (Sesia & Romano, 2021) could detect a possible asymmetry of the ground-truth distribution. However, our densities are estimated through classification techniques, whereas their densities are learned through a histogram of many regressors. Smoothness over the distribution is encoded in our loss function, whereas they use a post hoc subinterval finding algorithm through linear programming to prevent many disjoint intervals. Our techniques have some overlap with ordinal regression and ordinal classification. Xu et al. (2023) discusses various risk categories (similar to coverage) for ordinal classification, while our work considers different score functions where coverage serves as the loss function. Lu et al. (2022) discuss the adaptation of APS style score functions to accommodate the ordinal structure of classes, which was subsequently utilised in (Sesia & Romano, 2021).

## 3  CP VIA REGRESSION-AS-CLASSIFICATION

### 3.1  CLASSIFICATION CONFORMAL PREDICTION

We aim to compute a conformity function that accurately predicts the appropriateness of a label for a specific data point. Given that the distribution function across labels can adopt diverse forms, such as being bimodal, heavy-tailed, or heteroscedastic, our approach must effectively factor in such shapes while upholding its coverage precision. A frequently employed technique involves using the conditional label density as a conformity function, leading to reliable results in the classification context. Typically, practitioners perform conformal prediction for classification with probability estimates from a Softmax neural network that covers $K$ output logits using cross-entropy loss. Let us denote the parametrized density

$$q_\theta(\cdot \mid x) = \text{softmax}(f_\theta(x)), \text{ where } \text{softmax}(v)_j = \frac{\exp(v_j)}{\sum_{k=1}^{K} \exp(v_k)},$$

as the outputted discrete probability distribution over the labels of the input $x$. Traditionally, we fit our neural network by minimizing the cross-entropy loss on the training set:

$$\hat{\theta} \in \arg\min_\theta \sum_{i=1}^{n} \text{KL}(\delta_{y_i} \mid\mid q_\theta(\cdot \mid x_i)).$$

Here, $\delta_{y_i}$ is the Dirac Distribution with all of the probabilitistic mass on $y_i$. Let us assume that we have trained and acquired a $\hat{\theta}$ that has minimized the traditional cross-entropy loss function on the training dataset. A natural conformity score is simply the probability of a label according to the learned conditional distribution, i.e., $\sigma(x, y) = q_{\hat{\theta}}(y \mid x)$. This approach is both straightforward and efficient. The neural network's flexibility allows it to learn numerous label distributions with adaptivity across examples with less explicit design of specific prior structure.

## 3.2 Regression to Classification Approach

Naturally, we strive to embody such practicality and effectiveness in regression settings. However, the distribution of labels in the regression scenario is continuous, and learning a continuous distribution directly using a neural network is challenging (Rothfuss et al., 2019). Often, Bayesian or kernel density estimators are employed to estimate this distribution. Other techniques acquire knowledge of this distribution by training numerous regressors and categorizing the regressors, for example, Conformal Histogram Regression (Sesia & Romano, 2021).

However, these methods look different from the conformal prediction approaches for classification. It would be desirable to be able to use similar methods for both classification and regression conformal prediction. One method of unifying classification and regression problems outside the conformal prediction literature is known as *Regression-as-Classification*. We simply turn a regression problem into a classification problem by binning the range space. Specifically, we generate $K$ bins with $K$ equally spaced numbers covering the interval $\mathcal{Y} = [y_{\min}, y_{\max}]$, where $y_{\min}$ (or $y_{\max}$) is the minimum (or maximum) value of the labels observed in the training set. More explicitly, we define our discretization of the label space as

$$\hat{\mathcal{Y}} = \{\hat{y}_1, \ldots, \hat{y}_K\} \text{ where } \hat{y}_{k+1} = \hat{y}_k + \frac{\hat{y}_K - \hat{y}_1}{K - 1} \text{ with } \hat{y}_1 = y_{\min} \text{ and } \hat{y}_K = y_{\max}.$$

These values $\hat{y} \in \hat{\mathcal{Y}}$ form the midpoints for each bin of our discretization. Naturally, the $k$th bin is all the labels in the range space $\mathcal{Y}$ closest to $\hat{y}_k$. Intuitively, we can treat each bin as a class. Thus, we have turned a regression problem into a classification problem. This method is simple but has yielded surprising results. Some work has suggested that this form of binning results in more stable training (Stewart et al., 2023) and gives significantly better results for learning conditional expectations. To unify classification and regression conformal prediction, a simple solution is to employ the Classification Conformal Prediction model with discrete labels $\tilde{y}_i = \arg\min_{\hat{y} \in \hat{\mathcal{Y}}} |y_i - \hat{y}|$. This will aid in training the neural network with modified labels through cross-entropy loss, resulting in a discrete distribution of $q_\theta(\cdot \mid x)$, as outlined in the previous section.

To compute conformity scores for all labels, we employ linear interpolation from the discrete probability function $q_\theta(\cdot \mid x)$ to generate the continuous distribution $\bar{q}_\theta(\cdot \mid x)$. Nevertheless, this approach encounters a critical issue when employed for regression.

## 3.3 Data Fitting

A critical problem with employing CrossEntropy loss in the classification conformal prediction context is that *any structural relationships between classes are disregarded*. This is not surprising given that in the classification context, no structure exists between classes, and each class is independent. Therefore, the CrossEntropy loss does not need to differentiate between whether $q_\theta$ allocates probable mass far away or close to the actual label. Instead, it only incentivizes the allocation of probable mass on the correct label. However, within the regression setting, despite a multitude of labels, the labels adhere to an ordinal structure. Thus, to enhance the accuracy of labeling, it is imperative to devise a loss function that incentivizes the allocation of probabilistic mass not only to the correct bin but also to the neighboring bins. Formally, given an input and output pair $(x, y)$, our goal is to determine a density estimate $q_\theta(\cdot \mid x)$ that assigns low (resp. high) probability to points that are far (resp. close) to the true label $y$, i.e.,

$$q_\theta(\hat{y} \mid x_i) \text{ high (resp. small) when the loss } \ell(\hat{y}, y_i) \text{ is small (resp. high).}$$

Hence, a natural desideratum for learning the probability density function $q_\theta$ is that their product $\ell(y, \hat{y})q(\hat{y} \mid x)$ is small in expectation. We propose to find a distribution $q_\theta$ minimizing the loss

$$\mathbb{E}_{\hat{y} \sim q(\cdot \mid x)}[\ell(y, \hat{y})] = \sum_{k=1}^{K} \ell(y, \hat{y}_k)q(\hat{y}_k \mid x)$$

Minimizing this loss in the space of all possible distributions $q(\cdot \mid x)$ is equivalent to minimizing the original loss $\ell(y, \cdot)$, where the minimizing distribution is a Dirac delta $\delta_{\hat{y}_\star}$ at the minimizer $\hat{y}_\star$ of $\ell(y, \cdot)$. Therefore, we expect an unregularized version of this loss to share similarities with the typical empirical risk minimization on $\ell(y, \cdot)$. However, a key difference is that when minimizing in a restricted family of distributions (for example, those representable by a neural network with a fixed architecture), the distributional output can represent multi-modal or heavy-tailed label distributions. Minimizing the original loss $\ell(y, \cdot)$, one would always be confined to a point estimate.

## 3.4 Entropy Regularization

Although the proposed loss function better encodes the connection between bins, it tends towards outputting Dirac distributions, as outlined in the previous paragraph. Overconfidence in our neural network is a commonly reported problem in the literature (Wei et al., 2022). Nevertheless, smoothness has been a traditional requirement in density learning. As such, we rely on a classical entropy-regularization technique for learning density estimators (Wainwright & Jordan, 2008). Given the set of density estimators that match the training label distribution well and put high probability mass on the best bins, we prefer the probability distribution that maximizes the entropy since this intuitively takes fewer assumptions on the data distribution structure. Our choice is based on selecting density estimators that effectively match the distribution of the training labels and assign a higher probability to the best bins. Formally, we can calculate the entropy of our probability distribution by using the Shannon entropy $\mathcal{H}$ of the produced probability distribution $q(\cdot \mid x)$ as a penalty term as follows:

$$\mathcal{H}(q_\theta(\cdot \mid x)) = \sum_{k=1}^{K} q_\theta(\hat{y}_k|x) \log q_\theta(\hat{y}_k|x).$$

**Summary** We learn a distribution by minimizing the following expected loss over a given training data $\mathcal{D}_{\text{tr}} = \{(x_1, y_1), \ldots, (x_{n_{\text{tr}}}, y_{n_{\text{tr}}})\}$ of size $n_{\text{tr}}$:

$$\mathcal{L}(\theta) = \sum_{i=1}^{n_{\text{tr}}} \sum_{k=1}^{K} \ell(y_i, \hat{y}_k) \, q_\theta(\hat{y}_k \mid x_i) - \tau \mathcal{H}(q_\theta(\cdot \mid x_i)), \tag{2}$$

In particular, we choose $\ell(y_i, \hat{y}_k)$ as $\ell(y_i, \hat{y}_k) = |y_i - \hat{y}_k|^p$ where $p > 0$ is a hyperparameter. This selection functions as a natural distance metric that meets all required objectives. As a result, we can employ a technique similar to the aforementioned Classification Conformal Prediction methods. Initially, we train a Softmax neural network $f_\theta$ with $K$ logits, grounded on the loss function Equation (2) on the training dataset. To calculate conformity scores for the calibration set, we utilize the linearly interpolated $\sigma(x, y) = \bar{q}_\theta(y \mid x)$ for a specific data point $(x, y)$. Namely, for any $y$ between $\hat{y}_k$ and $\hat{y}_{k+1}$, $\bar{q}_\theta(y \mid x)$ is defined as

$$\bar{q}_\theta(y \mid x) = \gamma_k q_\theta(\hat{y}_k \mid x) + (1 - \gamma_k)q_\theta(\hat{y}_{k+1} \mid x)$$

$$\text{where } \gamma_k = \frac{\hat{y}_{k+1} - y}{\hat{y}_{k+1} - \hat{y}_k}.$$

Complete details of this procedure are outlined in Algorithm 1.

## 4 Experiments

All code to run our method can be installed via

```
pip install r2ccp.
```

We investigate the empirical behavior of our `R2CCP` (Regression-to-Classification Conformal Prediction) method, which we have explained in detail in Algorithm 1. We have three sets of experiments. The first one is described in Section 4.1 and presents empirical evidence of the algorithm's ability to produce narrower intervals by utilizing label density characteristics, including

---

**Algorithm 1** **R**egression to **C**lassication **C**onformal **P**rediction (R2CCP).

---

1: **Input:**
- Dataset $\mathcal{D}_n = \{(x_1, y_1), \dots, (x_n, y_n)\}$ and new input $x_{n+1}$
- Desired confidence level $\alpha \in (0, 1)$

2: **Hyperparameters:** temperature $\tau > 0$, $p > 0$, number of bins $K > 1$

3: Discretize the output space $[y_{\min}, y_{\max}]$ into $K$ equidistant bins with midpoints $\{\hat{y}_1, \dots, \hat{y}_K\}$

4: Randomly split the dataset $\mathcal{D}_n$ in training $\mathcal{D}_{\mathrm{tr}}$ and calibration $\mathcal{D}_{\mathrm{cal}}$

5: Find a distribution $q_{\hat{\theta}}(\cdot \mid x)$ by (approximately) optimizing on the training set $\mathcal{D}_{\mathrm{tr}}$

$$\hat{\theta} \in \arg\min_{\theta \in \mathbb{R}^d} \sum_{i=1}^{n_{\mathrm{tr}}} \sum_{k=1}^{K} |y_i - \hat{y}_k|^p q_\theta(\hat{y}_k \mid x_i) - \tau \mathcal{H}(q_\theta(\cdot \mid x_i))$$

where $q_\theta(\hat{y}_k \mid x) = \mathrm{softmax}(f_\theta(x))_k$ for a model (e.g., neural net) $f_\theta : \mathbb{R}^d \to \mathbb{R}^K$.

6: $\mathcal{S} \leftarrow \{\bar{q}_{\hat{\theta}}(y \mid x) \text{ for } (x, y) \in \mathcal{D}_{\mathrm{cal}}\}$ *# $\bar{q}_{\hat{\theta}}(\cdot \mid x)$ is linear interpolation of softmax probabilities.*

7: $Q_\alpha(\mathcal{D}_{\mathrm{cal}}) \leftarrow \mathrm{quantile}(\mathcal{S}, \alpha)$

8: **return** Conformal Set $\Gamma^{(\alpha)}(x_{n+1}) = \{z \in \mathbb{R} \mid \bar{q}_{\hat{\theta}}(z \mid x_{n+1}) \geq Q_\alpha(\mathcal{D}_{\mathrm{cal}})\}$

---

heteroscedasticity, bimodality, or a combination. Section 4.2 demonstrates the effectiveness of our algorithm on synthetic and real data by comparing it with various benchmarks from the Conformal Prediction literature in terms of length and coverage. Section 4.3 evaluates the effect of different loss functions on the final learned densities and their impact on the intervals produced. All experiments were run over 5 different seeds at a coverage level of $90\%$, and the standard error over the experiments is reported in the subscript. We do not tune the hyperparameters and keep values of $K = 50$, $p = 0.5$, and $\tau = 0.2$ constant across all experiments. For all experiments, we report length, meaning the length of all the sets predicted, and coverage, the percent of instances where the true label is contained in the predicted intervals.

### 4.1 SPECIFIC CHARACTERISTICS OF LABEL DISTRIBUTION

**Heteroscedascity** We generate a toy dataset where the input is one-dimensional. It contains samples from the following distribution: $y \sim \mathcal{N}(0, |x|)$. The label distribution is heteroscedastic, meaning the variance of the labels changes as the input changes. In traditional Conformal Prediction literature, many existing algorithms fail to capture heteroscedasticity, resulting in wide intervals for inputs $x$ where the label distribution of $y$ has low variance. However, our learned algorithm can directly learn this relation and adjust the outputted probability distribution accordingly. Thus, we see that the lengths of the intervals will increase as the variance of the label distribution increases, which is desirable. We see this relation in Figure 1a. Moreover, we also use the dataset generated from (Lei & Wasserman, 2014) as discussed in Appendix D. This dataset exhibits heteroskedascity and bimodality as $X$ passes $-0.5$. We see that our learned method can adjust intervals accordingly to maintain coverage and length for all $X$. We plot this in Figure 1b. We plot how the intervals (grey) change as the data distribution (black) changes. As the variance of the labels increases as $x$ increases, the produced intervals adaptively get wider, taking advantage of the heteroscedasticity.

**Bimodality** We showcase our algorithm's capability to address labels with a bimodal distribution. Our bimodal dataset is generated by repeatedly (1) sampling two sets of random features that are close geometrically and (2) giving one set of features a label of 1 plus some Gaussian noise and giving the other a label of $-1$ plus some Gaussian noise. Therefore, our dataset is comprised of many similar data points with bimodally distributed labels. This bimodal distribution is particularly hard for many existing CP algorithms to solve since it requires outputting two disjoint conformal sets to achieve low length. CQR, for instance, cannot deal with this circumstance and will generate a conformal set that covers the entire range space. However, our method is flexible enough to assign a low probability to labels between $-1$ and 1, and our resulting conformal set will not include these intermediate labels. We see this in Figure 6d. In Figure 6d, our outputted probability distribution has two modes around labels $-1$ and 1 and assigns a low probability value to the valley between the modes.

## 4.2 Comparison to other Conformal Prediction Algorithms

The crucial criteria for assessing a Conformal Prediction algorithm consist of (1) coverage, representing the percentage of generated conformal sets that incorporate accurate labels, and (2) the length of the generated conformal sets. Our baseline techniques consist of the Kernel Density Estimator (KDE) as proposed by Lei & Wasserman (2014), alongside a conformity score shown by the estimated probability density. Furthermore, we take into consideration Fong & Holmes (2021) (CB), which employs the likelihood of a posterior distribution in their conformity function, and Sesia & Romano (2021) (CHR), which uses quantile regressors on every bin of a histogram density estimator. We have included the Conformal Quantile Regression as described by Romano et al. (2019) (CQR), which employs conformity based on the labels' distance from quantile regressors. Moreover, we had the Distributional Conformal Prediction which uses a Cumulative Distribution Function to form its intervals from Chernozhukov et al. (2021) (DCP). Additionally, we have used the Lasso Conformal Predictor with a distance to mean regressors, which is the most straightforward option (LASSO). We use synthetic data exhibiting bimodally distributed and log-normally distributed to illustrate particular weaknesses of existing methods. We use real datasets also in Romano et al. (2019). Specifically, these are several datasets from the UCI Machine Learning repository (Bio, Blog, Concrete, Community, Energy, Forest, Stock, Cancer, Solar, Parkinsons, Pendulum) (Nottingham et al., 2023) and the medical expenditure panel survey number 19–21 (MEPS-19–21) (Cohen et al., 2009). These regression datasets are commonly used to benchmark regression models. Our approach yields tighter intervals on real datasets than some of the strongest baselines.

**Results** We report lengths and coverages results in Table 1 respectively. We added figures depicting example probability distributions on these datasets in Figure 6 in Appendix H. The intervals produced by our method are the shortest over 10 of the 16 datasets. From Figure 6, we see that our method can learn many different shaped distributions well, which accounts for this significant improvement in intervals. Overall, the Kernel Density Estimation and our method can accurately predict the best intervals on datasets where the connection between the data and feature is simple, such as Bimodal or Log-Normal. While the Kernel Density Estimator can fail when the labels are complexly related to inputs, no such connection exists in these datasets since the labels are independent of the features. Thus, the Kernel Density Method's simplicity allows it to learn the label density quickly. Our method also seems to handle the case where there is no connection between the data and the labels, as seen in Figure 6d in Appendix H. Moreover, on datasets where the label density is smooth and close to the Laplace prior, such as on Figure 6c and Figure 6i, Conformal Bayes and our method can accurately learn this distribution since both methods can output smooth distributions. Moreover, on very sharp datasets such as on Concrete in Figure 6j and Energy in Figure 6h, both our method and CQR can capture the sharp and unnoisy distribution needed to achieve strong length. Moreover, on complex distributions such as in Pendulum in Figure 6e and Bio in Figure 6f, we see that both CHR and our method have the flexibility to portray complex distributions resulting in the most accurate intervals. Thus, the flexibility of our algorithm to smoothly learn sharp, wide, complex, and simple conditional label densities results in our method achieving the best length most consistently over the entire dataset.

## 4.3 Ablation Studies

Our loss function consists of an error and entropy terms. The error term penalizes distributions that put weight far away from the true label, whereas the entropy term acts as a regularizing term in the probability distribution space. We will do ablations on both the error and entropy terms to illustrate the importance of each part. For the error term, there are several notable alternatives that a practitioner may use. An alternative is the log maximum likelihood or cross-entropy formulations of the error term, which we denote as

$$\mathcal{L}_{\text{MLE}}(\theta) = -\sum_{i=1}^{N} \log(q_\theta(\tilde{y}_i \mid x_i)),$$

where we remind the reader that $\tilde{y}_i = \arg\min_{\hat{y} \in \hat{y}} |\hat{y} - y_i|$. We note that both MLE and CE are equivalent formulations in this setting. These two are standard error terms often used in practice. We will train our models with this loss function over all datasets to see how the change of error term affects the intervals' length and the learned density. We will show that our chosen error term is better

| DATASET | BIMODAL | LOGNORM | CONCRETE | MEPS-19 | MEPS-20 | MEPS-21 | BIO | COMMUNITY |
|---|---|---|---|---|---|---|---|---|
| CQR | $2.14_{(0.01)}$ | $1.58_{(0.03)}$ | $\mathbf{0.39_{(0.01)}}$ | $1.87_{(0.05)}$ | $2.00_{(0.07)}$ | $1.99_{(0.03)}$ | $1.34_{(0.01)}$ | $\mathbf{1.44_{(0.03)}}$ |
| KDE | $\mathbf{0.35_{(0.01)}}$ | $\mathbf{1.40_{(0.04)}}$ | $1.54_{(0.03)}$ | $2.16_{(0.02)}$ | $2.51_{(0.05)}$ | $2.39_{(0.03)}$ | $2.27_{(0.00)}$ | $2.23_{(0.09)}$ |
| LASSO | $2.14_{(0.00)}$ | $3.30_{(0.06)}$ | $2.74_{(0.03)}$ | $4.64_{(0.06)}$ | $4.79_{(0.04)}$ | $4.92_{(0.04)}$ | $3.89_{(0.00)}$ | $3.26_{(0.05)}$ |
| CB | $2.16_{(0.01)}$ | $1.45_{(0.05)}$ | $0.96_{(0.01)}$ | $4.47_{(0.02)}$ | $4.50_{(0.02)}$ | $4.51_{(0.01)}$ | $2.09_{(0.00)}$ | $1.80_{(0.00)}$ |
| CHR | $2.14_{(0.00)}$ | $1.52_{(0.05)}$ | $0.47_{(0.02)}$ | $2.60_{(0.08)}$ | $2.53_{(0.03)}$ | $2.75_{(0.03)}$ | $1.59_{(0.01)}$ | $\mathbf{1.49_{(0.05)}}$ |
| DCP | $2.14_{(0.01)}$ | $1.74_{(0.07)}$ | $0.47_{(0.01)}$ | $68.64_{(0.15)}$ | $66.71_{(0.40)}$ | $67.56_{(0.33)}$ | $1.74_{(0.01)}$ | $1.59_{(0.03)}$ |
| R2CCP (OURS) | $0.46_{(0.01)}$ | $1.96_{(0.03)}$ | $\mathbf{0.38_{(0.01)}}$ | $\mathbf{1.60_{(0.01)}}$ | $\mathbf{1.70_{(0.03)}}$ | $\mathbf{1.72_{(0.03)}}$ | $\mathbf{1.11_{(0.01)}}$ | $1.47_{(0.03)}$ |

| DATASET | DIABETES | SOLAR | PARKINSONS | STOCK | CANCER | PENDULUM | ENERGY | FOREST |
|---|---|---|---|---|---|---|---|---|
| CQR | $1.30_{(0.05)}$ | $1.98_{(0.22)}$ | $\mathbf{0.42_{(0.01)}}$ | $1.85_{(0.23)}$ | $\mathbf{3.09_{(0.13)}}$ | $2.25_{(0.31)}$ | $\mathbf{0.19_{(0.01)}}$ | $3.18_{(0.19)}$ |
| KDE | $1.34_{(0.05)}$ | $\mathbf{0.50_{(0.01)}}$ | $3.79_{(0.02)}$ | $4.72_{(0.29)}$ | $3.82_{(0.09)}$ | $3.96_{(0.09)}$ | $2.72_{(0.07)}$ | $\mathbf{2.90_{(0.17)}}$ |
| LASSO | $3.01_{(0.04)}$ | $3.54_{(0.12)}$ | $3.46_{(0.03)}$ | $1.39_{(0.04)}$ | $3.55_{(0.14)}$ | $3.99_{(0.07)}$ | $1.29_{(0.03)}$ | $3.97_{(0.29)}$ |
| CB | $\mathbf{1.19_{(0.01)}}$ | $3.78_{(0.02)}$ | $3.42_{(0.01)}$ | $1.32_{(0.01)}$ | $\mathbf{3.14_{(0.04)}}$ | $3.71_{(0.03)}$ | $1.26_{(0.01)}$ | $3.75_{(0.03)}$ |
| CHR | $1.40_{(0.02)}$ | $1.49_{(0.23)}$ | $0.68_{(0.02)}$ | $1.59_{(0.07)}$ | $3.42_{(0.12)}$ | $\mathbf{1.69_{(0.11)}}$ | $0.23_{(0.01)}$ | $\mathbf{3.03_{(0.15)}}$ |
| DCP | $1.29_{(0.02)}$ | $15.69_{(0.06)}$ | $0.83_{(0.04)}$ | $1.69_{(0.10)}$ | $3.57_{(0.07)}$ | $1.76_{(0.10)}$ | $0.23_{(0.01)}$ | $6.00_{(0.02)}$ |
| R2CCP (OURS) | $1.34_{(0.02)}$ | $3.80_{(2.61)}$ | $0.50_{(0.00)}$ | $\mathbf{0.92_{(0.02)}}$ | $3.21_{(0.08)}$ | $\mathbf{1.60_{(0.07)}}$ | $0.20_{(0.02)}$ | $3.80_{(0.26)}$ |

Table 1: This is the length results over all datasets. We see that our method achieves the best length on 10 of the 16 datasets. Meanwhile, CQR is best at 5, CHR is best at 3, CB is best at 1, and KDE is the best at 3. Our method achieves the shortest intervals across these datasets.

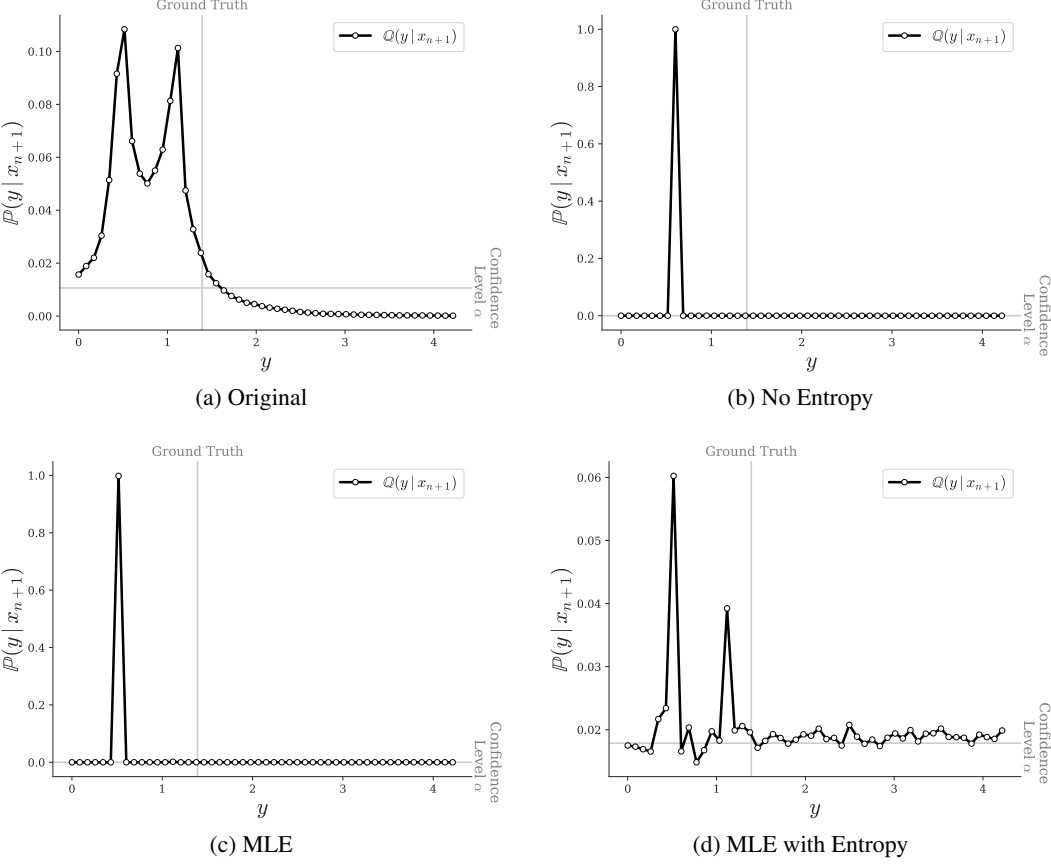

Figure 2: The resulting density estimates with different loss functions. We see that removing entropy from our loss function or using MLE as error terms causes sharp density estimates. Moreover, adding in the entropy regularization with MLE does not smooth the density estimate but instead raises the entire distribution uniformly; which does not provide valuable information for CP.

for producing optimal intervals empirically. Moreover, we will also demonstrate the importance of our entropy term. We do this by retraining our models with the entropy part omitted. We also test by combining the MLE error term with entropy regularization. Therefore, we will perform 4 different

| DATASET | BIMODAL | LOG-NORMAL | CONCRETE | MEPS-19 | MEPS-20 | MEPS-21 | BIO | COMMUNITY |
|---|---|---|---|---|---|---|---|---|
| $\mathcal{L}_{\text{NE}}$ | $0.43_{0.001}$ | $3.25_{0.058}$ | $1.89_{0.037}$ | $3.73_{0.057}$ | $3.79_{0.061}$ | $8.78_{0.163}$ | $2.01_{0.006}$ | $3.64_{0.054}$ |
| $\mathcal{L}_{\text{MLE}}$ | $\mathbf{0.35_{0.001}}$ | $\mathbf{1.76_{0.020}}$ | $0.74_{0.016}$ | $1.58_{0.008}$ | $1.59_{0.009}$ | $1.71_{0.034}$ | $2.71_{0.001}$ | $2.32_{0.047}$ |
| $\mathcal{L}_{\text{MLE + E}}$ | $0.36_{0.001}$ | $3.52_{0.008}$ | $1.91_{0.017}$ | $1.60_{0.008}$ | $1.63_{0.027}$ | $\mathbf{1.70_{0.008}}$ | $2.32_{0.003}$ | $3.77_{0.013}$ |
| $\mathcal{L}$ | $0.44_{0.002}$ | $1.82_{0.034}$ | $\mathbf{0.37_{0.004}}$ | $\mathbf{1.60_{0.009}}$ | $\mathbf{1.59_{0.011}}$ | $1.69_{0.013}$ | $\mathbf{1.10_{0.004}}$ | $\mathbf{1.50_{0.025}}$ |

| DATASET | DIABETES | SOLAR | PARKINSONS | STOCK | CANCER | PENDULUM | ENERGY | FOREST |
|---|---|---|---|---|---|---|---|---|
| $\mathcal{L}_{\text{NE}}$ | $1.92_{0.052}$ | $2.20_{0.233}$ | $4.76_{0.033}$ | $10.17_{0.165}$ | $3.26_{0.207}$ | $12.82_{0.479}$ | $3.30_{0.093}$ | $3.33_{0.254}$ |
| $\mathcal{L}_{\text{MLE}}$ | $1.56_{0.031}$ | $\mathbf{0.14_{0.005}}$ | $0.34_{0.006}$ | $4.48_{0.214}$ | $3.26_{0.117}$ | $3.00_{0.203}$ | $0.25_{0.011}$ | $3.04_{0.110}$ |
| $\mathcal{L}_{\text{MLE + E}}$ | $1.96_{0.012}$ | $0.15_{0.005}$ | $\mathbf{0.23_{0.001}}$ | $9.73_{0.074}$ | $3.95_{0.022}$ | $13.40_{0.221}$ | $0.25_{0.009}$ | $5.24_{0.055}$ |
| $\mathcal{L}$ | $\mathbf{1.37_{0.037}}$ | $0.66_{0.092}$ | $0.46_{0.036}$ | $\mathbf{1.95_{0.039}}$ | $\mathbf{3.04_{0.142}}$ | $\mathbf{1.62_{0.042}}$ | $\mathbf{0.21_{0.025}}$ | $\mathbf{2.92_{0.127}}$ |

Table 2: We present the length results over all of the variant loss functions. We find that our loss function delivers the best over 12 datasets, demonstrating that our chosen loss function often generates the best intervals. For datasets where our method does not deliver the best results, it is likely that tuning the weight on the entropy $\tau$ and the smoothing term $p$ would likely have improved the results, but we do not do this for the sake of evaluation.

ablation experiments on the loss function by retraining on different variations of the loss functions and reporting the final length and coverage generated. Overall, the four loss functions we will use are the original loss function $\mathcal{L}$, the original loss function without entropy $\mathcal{L}_{\text{NE}}$, the MLE loss function $\mathcal{L}_{\text{MLE}}$, and the MLE loss function with entropy added $\mathcal{L}_{\text{MLE + E}}$. We will demonstrate that our chosen loss function delivers the best results across all the loss functions. We report our results in Table 2.

**Results**   We see that our chosen loss function achieves the best length across most loss functions in Table 2. The only datasets where our chosen loss function does not achieve the best length are Energy, Solar, Bimodal, and Log-Normal. We now discuss the individual differences between our loss function and each variant loss function. When comparing our loss $\mathcal{L}$ with the variant without entropy $\mathcal{L}_{\text{NE}}$, we see the lengths constantly increase except for the bimodal distribution. Without an entropy regularizing term, the outputted probability distributions are not smooth, placing much mass on a single data point. This overconfidence results in the $\alpha$th quantile of the low probability of the true label as in Figure 2.

When looking at the differences between our loss function when changing the error term to Maximum Log Likelihood, we see similar overconfidence. While MLE loss works on several datasets, such as Energy, Solar, Bimodal, and Log-Normal, it also performs poorly on other datasets. MLE works well for datasets with less simple label distributions but fails otherwise due to similar overconfidence. Even when adding entropy as a regularizer to the MLE loss as in $\mathcal{L}_{\text{MLE + E}}$, we see that the addition of entropy does not improve the length. Since MLE loss does not treat nearby bins differently than far away bins, regularization decreases the overconfidence but does so uniformly across all bins. The result is a roughly uniform distribution with a single spike. This does not utilize the structure of regression where bins near the best bin are preferable. Thus, prioritizing nearby bins is crucial for achieving strong length. Moreover, when adding entropy regularization to our error term, we do not see a uniform increase in probability across all bins but instead a smoothening of a direct. This connection between entropy regularization and our distance-based loss function appears to be a powerful synergy per the intervals produced. Thus, the error term utilizing the regression structure by prioritizing nearby bins and the regularizing entropy term preventing overconfidence seem crucial for producing good intervals.

Another important insight is that the datasets where other loss functions excel are the same ones where other CP methods perform better than ours. In particular, on the Bimodal, Log-Normal, Parkinson's, and Solar datasets, our CP method is not superior, and other loss functions outperform it. This indicates that the MLE's sharp distributions result in greater accuracy on these specific datasets. Hence, adjusting the weighting terms for entropy $\tau$ and smoothing $p$ could enhance the smoothness of the learned distribution and, thereby, the performance of our algorithm across these datasets, leading to the best outcomes. Nevertheless, to maintain fairness, we avoid this approach.

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

## A  AUTHOR CONTRIBUTION STATEMENT

Etash Guha, Shlok Natarajan, Thomas Möllenhoff, Eugene Ndiaye, and Mohammad Emtiyaz Khan were all responsible for the development of the main idea and writing of the paper. Etash Guha and Shlok Natarjan were responsible for the experiments and implementation of the algorithm. Etash Guha, Thomas Möllenhoff, and Eugene Ndiaye were responsible for the design of the loss function, binning, and entropic regularization. Etash Guha and Eugene Ndiaye were responsible for writing the related works.

## B  LIMITATIONS

There are several limitations to our algorithm. The loss function we choose to optimize needs larger representational power due to increased complexity of the output. Therefore, the neural networks we used in practice to minimize this loss function are larger than the ones used for CQR. Moreover, to achieve strong training, we had to slightly vary the hyperparameters for a dataset depending on its size. For example, larger datasets needed models to be trained with smaller initial learning rate to avoid divergence. This was not needed for Quantie Regressors. If the network does not train well or achieve good training or validation loss, the intervals produced will be suboptimal, so finding a training setup that minimizes the loss effectively is crucial for our algorithm to produce meaningful intervals. Moreover, we do not directly learn the label distribution and our produced probability distributions do not directly mirror that of the ground truth distribution. One reason for this is the use of entropy regularization. While using entropy regularization introduces bias, we found it to be necessary to prevent the neural network from being overconfident on a simple bin in practice.

## C  MORE DETAILS ON CONFORMAL PREDICTION AND VARIANTS

Given an input observation $x$ without its label $y$, Conformal Prediction method Vovk et al. (2005) allows to select set of values that its unobserved response $y$ are likely to take. For an arbitrary value $z$, the rational is to score how the example $(x_{n+1}, z)$ is a typical/conformal observation given a sequence of iid data $\mathcal{D}_n = \{(x_i, y_i)\}_{i \in [n]}$. Intuitively, the scoring function, that we denote $\sigma$, measures how different is the new example compared to examples that we have already observed. We compute the score function $\sigma(x, y)$ for every example in the augmented dataset $\mathcal{D}_{n+1}(z) = \mathcal{D}_n \cup (x, z)$ and report the fraction of time previous example score better than the new one.

This define the so called *conformity function* $\pi(\cdot) \in [0, 1]$ as

$$\pi(z) = 1 - \frac{\text{Rank}(z)}{n+1} \text{ where } \text{Rank}(z) = \sum_{(x,y) \in D_{n+1}(z)} \mathbb{I}\{\sigma(x, y) \leq \sigma(x_{n+1}, z)\}. \tag{3}$$

If $\pi(z)$ is small (resp. large) then $z$ is non conformal (resp. conformal). Therefore, for a threshold level $\alpha \in (0, 1)$, a simple way to find our conformal set $\Gamma^{(\alpha)}(x_{n+1})$ for the response $y_{n+1}$ of the input $x_{n+1}$ is to select all the values $z$ whose conformity exceed the threshold i.e.

$$\Gamma^{(\alpha)}(x_{n+1}) = \{z \in \mathbb{R} : \pi(z) \geq \alpha\}.$$

The validity of the method rely on the fact that if we assume that order in which the observations arrive is irrelevant and that the score function $\sigma$ is invariant wrt permutation of the data, then the sequence of scores $\{\sigma(x_1, y_1), \ldots, \sigma(x_n, y_n), \sigma(x_{n+1}, y_{n+1})\}$ are also equally likely and thus the $\text{Rank}(y_{n+1})$ is uniformly distributed in $\{1, \ldots, n+1\}$. As such, for any confidence level $\alpha \in (0, 1)$,

$$\mathbb{P}(y_{n+1} \in \Gamma^{(\alpha)}(x_{n+1})) \geq \mathbb{P}(\pi(y_{n+1}) \geq \alpha) = \mathbb{P}(\text{Rank}(y_{n+1}) \leq (n+1)(1-\alpha)) \geq 1 - \alpha.$$

**Conditional validity**   The previous coverage guarantee may not always be ideal because it is not conditional on the particular input $x_{n+1}$. Unfortunately, it is not possible to have such conditional validity when the distribution of the inputs is continuous. Nevertheless, it can be approximated using some localisation strategies based on a weighted conformity function; see (Lin et al., 2021; Ghosh et al., 2023). This also allows to further reduce the size of the conformal prediction sets.

Validity can be conditional on a discrete attribute of the input, $x_{n+1}$, through the use of Mondrian Conformal Prediction. This method involves a slight modification of the conformity function. Let

a taxonomy $\kappa$ be a function that partitions the data into a finite number of categories eg $\kappa(x,.)$ is a classification of the example $x$. This allows to modify the pi-value function in order to achieve validity conditional on the category of the test example :

$$\pi(z) = \sum_{(x,y) \in \mathcal{D}_{n+1}(z)} \frac{\mathbb{I}\{\sigma(x,y) \leq \sigma(x_{n+1}, z), \kappa(x,y) = \kappa(x_{n+1}, z)\}}{\sum_{(x,y) \in \mathcal{D}_{n+1}(z)} \mathbb{I}\{\kappa(x,y) = \kappa(x_{n+1}, z)\}}$$

**Computational issues**   As mentioned in the introduction, the selection of the score function affects the size and shape of the prediction sets. There are also computational issues that arise when the full data set is used to fit the function. For example, consider using the estimated distance to the conditional mean $\sigma(x,y) = |y - \hat{E}[y|x]|$. The estimator $\hat{E}[\cdot|x]$ is learned on the extended dataset $\mathcal{D}_{n+1}(z)$ for all possible values $z$, which is computationally infeasible for most methods without stronger assumptions; see (Ndiaye & Takeuchi, 2023), (Ndiaye, 2022). As such, most of the popular strategies are based on the fitting of the prediction model to an left-out independent set of data, as we did in this paper.

## D   DETAILS ON DATASET FROM LEI & WASSERMAN (2014)

For the dataset referenced in Figure 1b, we generate the dataset by sampling many $(X, Y)$ pairs from the following distribution.

$$X \sim \text{Unif}[-1.5, 1.5]$$
$$Y \mid X \sim \frac{1}{2}\mathcal{N}(f(X) - g(X), \sigma^2(X)) + \frac{1}{2}\mathcal{N}(f(X) + g(X), \sigma^2(X))$$
$$f(X) = (X-1)^2(X+1), \; g(X) = 4\sqrt{(X+1/2)\mathbb{I}(X \geq -1/2)}, \; \sigma^2(X) = 1/4 + |X|$$

This dataset demonstrates bimodality after $x$ passes the threshold of $-0.5$. This dataset was similarly used by the Lei & Wasserman (2014).

## E   COVERAGE DATA

We have presented the coverage data for the Conformal Prediction comparison as well as the ablation experiments in Table 3 and Table 4, respectively. Since all methods obey the classical Conformal Prediction framework, we expect coverage at the guaranteed level of $1 - \alpha$. This guarantee is indeed what we see across all experiments. This confirms that our coverage guarantee indeed holds in practice.

| DATASET | BIMODAL | LOGNORM | CONCRETE | MEPS-19 | MEPS-20 | MEPS-21 | BIO | COMMUNITY |
|---|---|---|---|---|---|---|---|---|
| CQR | $0.90_{(0.00)}$ | $0.91_{(0.00)}$ | $0.91_{(0.00)}$ | $0.90_{(0.00)}$ | $0.90_{(0.00)}$ | $0.90_{(0.00)}$ | $0.90_{(0.00)}$ | $0.90_{(0.00)}$ |
| KDE | $0.93_{(0.00)}$ | $0.90_{(0.00)}$ | $0.90_{(0.00)}$ | $0.91_{(0.00)}$ | $0.92_{(0.00)}$ | $0.91_{(0.00)}$ | $0.90_{(0.00)}$ | $0.90_{(0.00)}$ |
| LASSO | $0.90_{(0.00)}$ | $0.91_{(0.00)}$ | $0.90_{(0.00)}$ | $0.90_{(0.00)}$ | $0.90_{(0.00)}$ | $0.90_{(0.00)}$ | $0.90_{(0.00)}$ | $0.90_{(0.00)}$ |
| CB | $0.90_{(0.00)}$ | $0.91_{(0.01)}$ | $0.88_{(0.01)}$ | $0.90_{(0.00)}$ | $0.90_{(0.00)}$ | $0.89_{(0.00)}$ | $0.90_{(0.00)}$ | $0.90_{(0.00)}$ |
| CHR | $0.90_{(0.00)}$ | $0.91_{(0.00)}$ | $0.91_{(0.00)}$ | $0.90_{(0.00)}$ | $0.90_{(0.00)}$ | $0.90_{(0.00)}$ | $0.90_{(0.00)}$ | $0.90_{(0.00)}$ |
| DCP | $0.90_{(0.00)}$ | $0.91_{(0.00)}$ | $0.91_{(0.00)}$ | $1.00_{(0.00)}$ | $1.00_{(0.00)}$ | $1.00_{(0.00)}$ | $1.00_{(0.00)}$ | $0.90_{(0.00)}$ |
| R2CCP (OURS) | $0.90_{(0.00)}$ | $0.90_{(0.00)}$ | $0.90_{(0.00)}$ | $0.90_{(0.00)}$ | $0.90_{(0.00)}$ | $0.90_{(0.00)}$ | $0.90_{(0.00)}$ | $0.90_{(0.00)}$ |

| DATASET | DIABETES | SOLAR | PARKINSONS | STOCK | CANCER | PENDULUM | ENERGY | FOREST |
|---|---|---|---|---|---|---|---|---|
| CQR | $0.91_{(0.00)}$ | $0.91_{(0.00)}$ | $0.90_{(0.00)}$ | $0.91_{(0.00)}$ | $0.92_{(0.00)}$ | $0.91_{(0.00)}$ | $0.90_{(0.00)}$ | $0.91_{(0.00)}$ |
| KDE | $0.91_{(0.00)}$ | $0.71_{(0.15)}$ | $0.89_{(0.00)}$ | $0.90_{(0.00)}$ | $0.92_{(0.01)}$ | $0.90_{(0.01)}$ | $0.91_{(0.00)}$ | $0.81_{(0.08)}$ |
| LASSO | $0.91_{(0.00)}$ | $0.90_{(0.00)}$ | $0.90_{(0.00)}$ | $0.91_{(0.00)}$ | $0.92_{(0.00)}$ | $0.90_{(0.00)}$ | $0.90_{(0.00)}$ | $0.90_{(0.00)}$ |
| CB | $0.89_{(0.00)}$ | $0.92_{(0.01)}$ | $0.90_{(0.00)}$ | $0.88_{(0.01)}$ | $0.87_{(0.02)}$ | $0.88_{(0.01)}$ | $0.89_{(0.01)}$ | $0.90_{(0.01)}$ |
| CHR | $0.91_{(0.00)}$ | $0.91_{(0.00)}$ | $0.90_{(0.00)}$ | $0.91_{(0.00)}$ | $0.92_{(0.00)}$ | $0.91_{(0.00)}$ | $0.91_{(0.00)}$ | $0.91_{(0.00)}$ |
| DCP | $0.91_{(0.00)}$ | $1.00_{(0.00)}$ | $0.90_{(0.00)}$ | $0.91_{(0.00)}$ | $0.92_{(0.00)}$ | $0.90_{(0.00)}$ | $0.91_{(0.00)}$ | $1.00_{(0.00)}$ |
| R2CCP (OURS) | $0.90_{(0.00)}$ | $0.92_{(0.02)}$ | $0.96_{(0.00)}$ | $0.92_{(0.02)}$ | $0.90_{(0.00)}$ | $0.90_{(0.00)}$ | $0.91_{(0.01)}$ | $0.89_{(0.00)}$ |

Table 3: This is the coverage data over all datasets. We see that all methods achieve the roughly $1 - \alpha$ coverage guaranteed by conformal prediction.

| DATASET | BIMODAL | LOGNORM | CONCRETE | MEPS-19 | MEPS-20 | MEPS-21 | BIO | COMMUNITY |
|---|---|---|---|---|---|---|---|---|
| $\mathcal{L}$ | $0.90_{(0.02)}$ | $0.90_{(0.02)}$ | $0.90_{(0.02)}$ | $0.90_{(0.01)}$ | $0.90_{(0.01)}$ | $0.90_{(0.01)}$ | $0.90_{(0.00)}$ | $0.90_{(0.01)}$ |
| $\mathcal{L}_{\text{NE}}$ | $0.90_{(0.02)}$ | $0.81_{(0.03)}$ | $0.84_{(0.03)}$ | $0.90_{(0.01)}$ | $0.90_{(0.01)}$ | $0.90_{(0.01)}$ | $0.68_{(0.01)}$ | $0.88_{(0.02)}$ |
| $\mathcal{L}_{\text{MLE}}$ | $0.90_{(0.02)}$ | $0.90_{(0.02)}$ | $0.90_{(0.02)}$ | $0.90_{(0.01)}$ | $0.90_{(0.01)}$ | $0.90_{(0.01)}$ | $0.90_{(0.00)}$ | $0.90_{(0.01)}$ |
| $\mathcal{L}_{\text{MLE + E}}$ | $0.90_{(0.02)}$ | $0.90_{(0.02)}$ | $0.90_{(0.02)}$ | $0.90_{(0.01)}$ | $0.90_{(0.01)}$ | $0.90_{(0.01)}$ | $0.92_{(0.00)}$ | $0.90_{(0.01)}$ |

| DATASET | DIABETES | SOLAR | PARKINSONS | STOCK | CANCER | PENDULUM | ENERGY | FOREST |
|---|---|---|---|---|---|---|---|---|
| $\mathcal{L}$ | $0.90_{(0.03)}$ | $0.90_{(0.02)}$ | $0.95_{(0.01)}$ | $0.90_{(0.03)}$ | $0.90_{(0.05)}$ | $0.90_{(0.03)}$ | $0.90_{(0.03)}$ | $0.89_{(0.03)}$ |
| $\mathcal{L}_{\text{NE}}$ | $0.94_{(0.02)}$ | $0.90_{(0.02)}$ | $0.73_{(0.01)}$ | $0.89_{(0.03)}$ | $0.90_{(0.05)}$ | $0.90_{(0.03)}$ | $0.90_{(0.02)}$ | $0.77_{(0.04)}$ |
| $\mathcal{L}_{\text{MLE}}$ | $0.90_{(0.03)}$ | $0.90_{(0.02)}$ | $0.90_{(0.01)}$ | $0.91_{(0.03)}$ | $0.90_{(0.05)}$ | $0.90_{(0.03)}$ | $0.90_{(0.02)}$ | $0.89_{(0.03)}$ |
| $\mathcal{L}_{\text{MLE + E}}$ | $0.90_{(0.03)}$ | $0.90_{(0.02)}$ | $0.90_{(0.01)}$ | $0.90_{(0.03)}$ | $0.90_{(0.05)}$ | $0.90_{(0.03)}$ | $0.90_{(0.03)}$ | $0.90_{(0.03)}$ |

Table 4: This is the coverage data for different loss functions from the ablation.

## F  MORE EXPERIMENTAL DETAILS

In order to maintain fairness across all baselines, we use the same size neural network for our method across all experiments. Specifically, we discretize the range space into $K = 50$ points, weight the entropy term by $\tau = 0.2$, use a 1000 hidden dimension, use 4 layers, use weight decay of $1e - 4$, use $p = .5$, and use AdamW as an optimizer. For most of the experiments, we use learning rate $1e - 4$ and batch size 32. However, for certain datasets, namely the MEPS datasets, we used a larger batch size of 256 to improve training time and used a smaller learning rate to prevent training divergence. We did change any other parameter between all of our runs. For the baselines of CQR and CHR, we use the neural network configurations mentioned in the paper. We found that the parameterizations mentioned by the authors in the papers achieved the best performance and changing the parameterizations weakened their results.

## G  ADDITIONAL EXPERIMENTS

We add several additional experiments to help contextualize our results. First, we add experiments tracking how our method performs over different coverage levels relative to CHR and CQR across several datasets in Appendix G.2. Second, in Appendix G.3, we analyze what percentage of the time the predicted intervals are singeltons, meaning there is only one continuous interval. Third, we analyze the correlation between the residual prediction error and the predicted interval's length with our method in Appendix G.4.

### G.1  PERFORMANCE WITH DIFFERENT NUMBER OF BINS

We set the number of bins at a constant $K = 50$. A natural question is how the number of bins affects the performance of our conformal prediction methodology. To test this, we retrain several models with varying number of bins on the Concrete dataset. We report the size of the predicted intervals for each number of bins in Figure 3. We observe that as the number of bins increases, the average length of the predicted intervals decreases and stabilizes. This suggests that after a sufficient number of bins, the length of the intervals becomes relatively constant. In practice, classification problems with 1000 bins or more have been routinely solved since 2012 (Krizhevsky et al., 2012). This suggests that the increasing number of bins does not make the learning task more difficult, explaining how performance remains relatively unaffected as the number of bins is increased.

### G.2  DIFFERENT COVERAGE LEVELS

While the coverage level of $\alpha = .1$ is relatively standard across the Conformal Prediction literature, we want to see how our method compares to other methods across different coverage levels. We take the trained models from the original experiments and compute the intervals across different coverage levels for our method, CHR, and CQR. As expected, for all methods, the length of the predicted intervals increases as the required coverage level increases. We report our results in Figure 4.

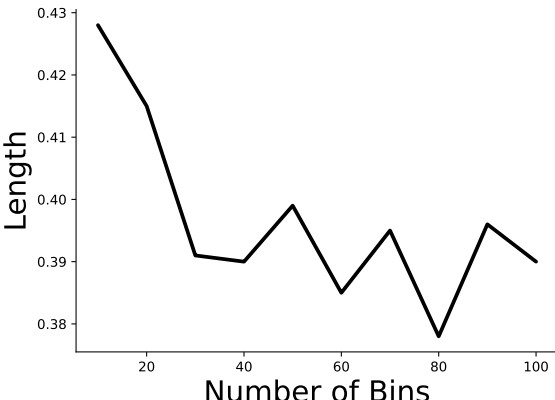

Figure 3: We present an ablation of how the number of bins affects the average length of the generated intervals.

### G.3 NUMBER OF SINGLETONS

We also provide an analysis of the percentage of times that our predicted intervals are singleton, meaning only one interval is predicted. We present our results in Table 5. We see that for a majority of the datasets, our model tends to predict singleton intervals. However, for both the Bimodal dataset and the Bio dataset, we see that our method predicts far less singletons. This observation is intuitive for the Bimodal dataset as two intervals are expected to account for the different modes.

### G.4 CORRELATION BETWEEN PREDICTION ERROR AND LENGTH OF PREDICTED INTERVALS

We want to see how the prediction error correlates with the predicted interval lengths. We define prediction error as the absolute error residual between predicted label and the true label. Given our model is taylored to generated intervals, there are many ways to predict a single label for a given set of features. For example, we can select the bin with the largest probability predicted by our model

$$\hat{y} = \hat{y}_l \text{ where } k = \arg\max_{k \in [K]} q_\theta(\hat{y}_k | x).$$

We can also use the expected label according to our distribution as the predicted label

$$\hat{y} = \sum_{k=1}^{K} \hat{y}_k q_\theta(\hat{y}_k | x).$$

We choose the second method for the sake of these experiments. We compute the absolute residual error as $\|\hat{y} - y\|$. We plot our results in Figure 5. We see that the prediction error is not strongly correlated with the length of the predicted interval.

## H OUTPUT DISTRIBUTION

Here, we plot the distribution of lengths of all Conformal Prediction methods over all datasets. We also plot example density functions learned by out method, CHR, and KDE on all datasets. We note that KDE's learned densities seem to be relatively less informative. Moreover, the learned densities from CHR are noisy and not smooth. Moreover, we plot several examples of only our learned probability distribution. We use this when referencing the experiments to demonstrate the different shapes of label distributions from the data.

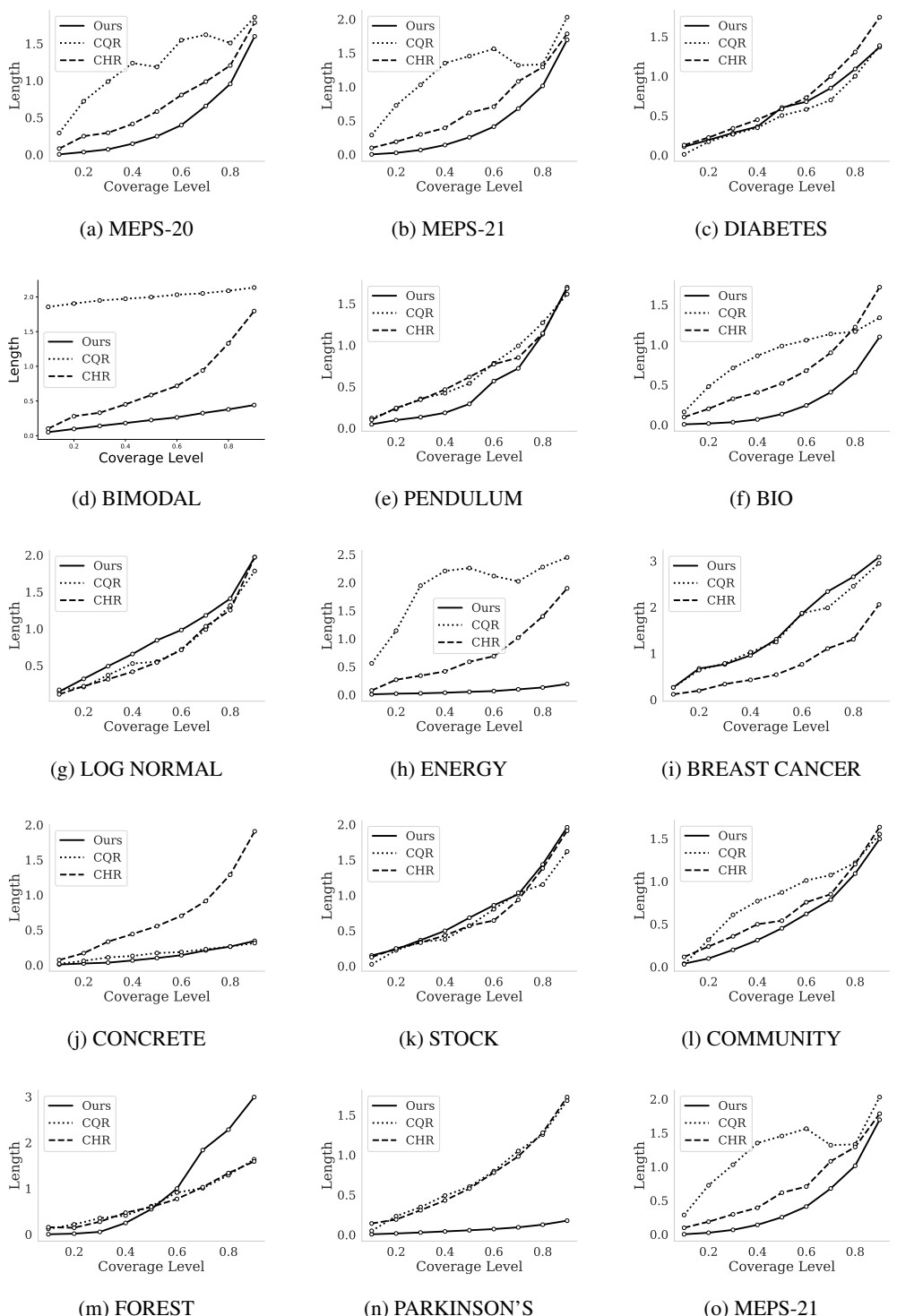

Figure 4: We plot the width of the predicted intervals for our method, CHR, and CQR across different coverage levels.

| DATASET | BIMODAL | LOG-NORMAL | CONCRETE | MEPS-19 | MEPS-20 | MEPS-21 | BIO | COMMUNITY |
|---|---|---|---|---|---|---|---|---|
| NUMBER OF SINGLETONS | 5 | 197 | 194 | 3157 | 3509 | 3132 | 6725 | 382 |
| NUMBER OF EXAMPLES | 400 | 200 | 206 | 3157 | 3509 | 3132 | 9146 | 399 |
| DATASET | DIABETES | SOLAR | PARKINSONS | STOCK | CANCER | PENDULUM | ENERGY | FOREST |
| NUMBER OF SINGLETONS | 81 | 174 | 1103 | 105 | 37 | 115 | 153 | 96 |
| NUMBER OF EXAMPLES | 89 | 214 | 1175 | 108 | 39 | 126 | 154 | 104 |

Table 5: We present an analysis of the number of singletons predicted by our method over all datasets. For most datasets, we see that our method tends to predict singletons.

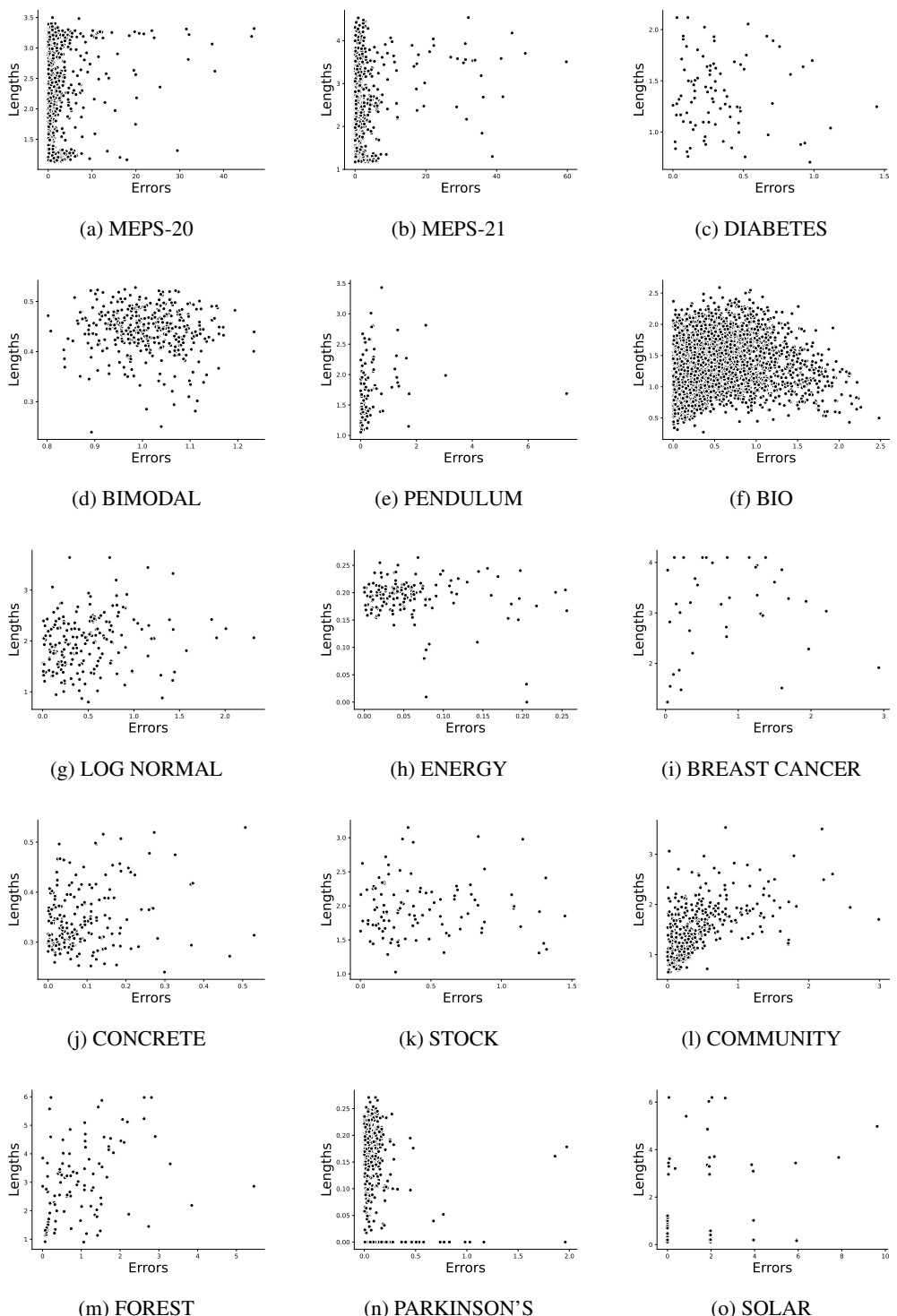

Figure 5: We plot the corrleation between absolute residual error and the length of our predicted intervals over several datasets. We see that there is not a strong correlation.

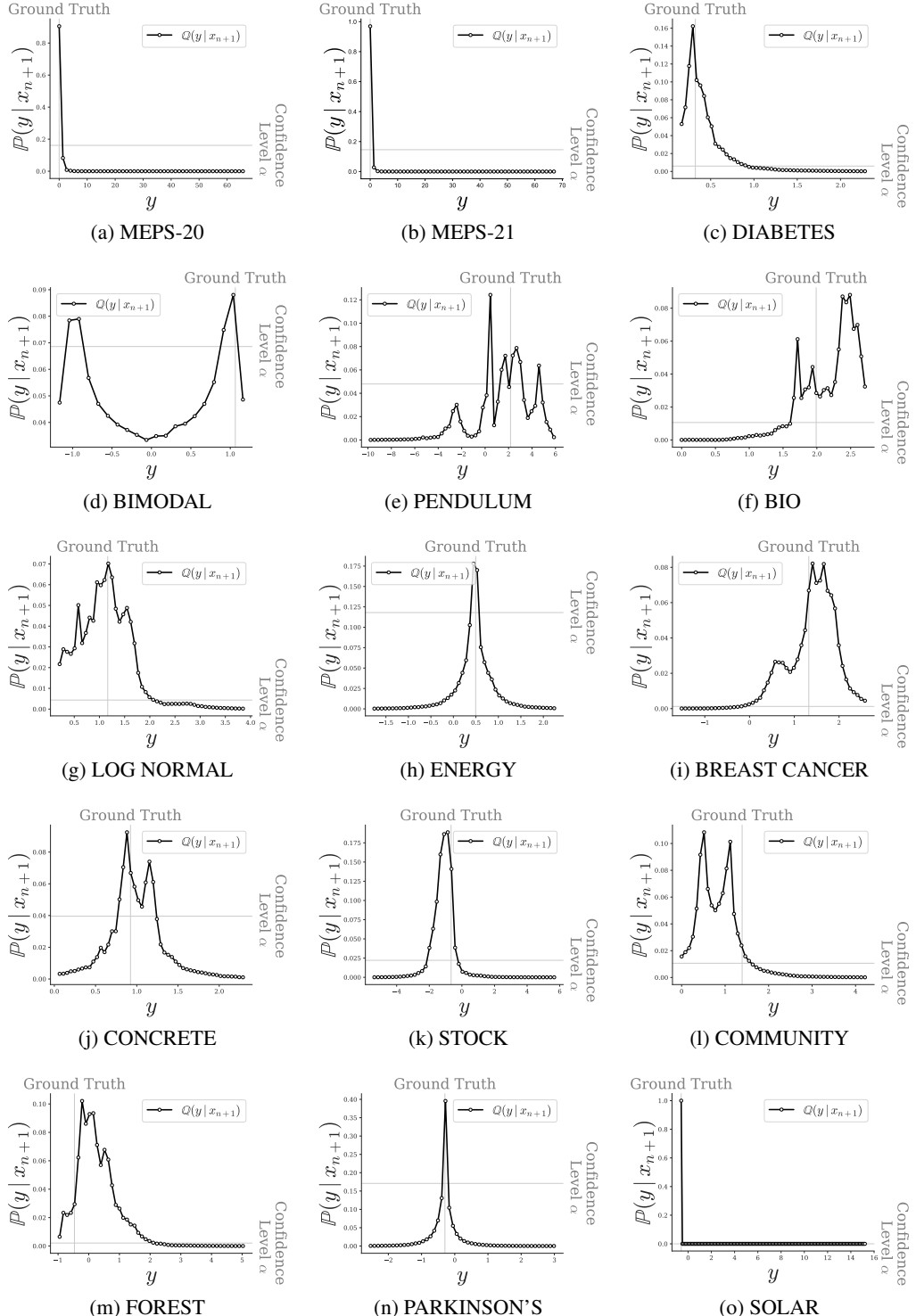

Figure 6: Example Probability Distributions outputted by our learned methodology on different datapoints from different datasets. We can see there are a variety of different shapes of distributions learned.

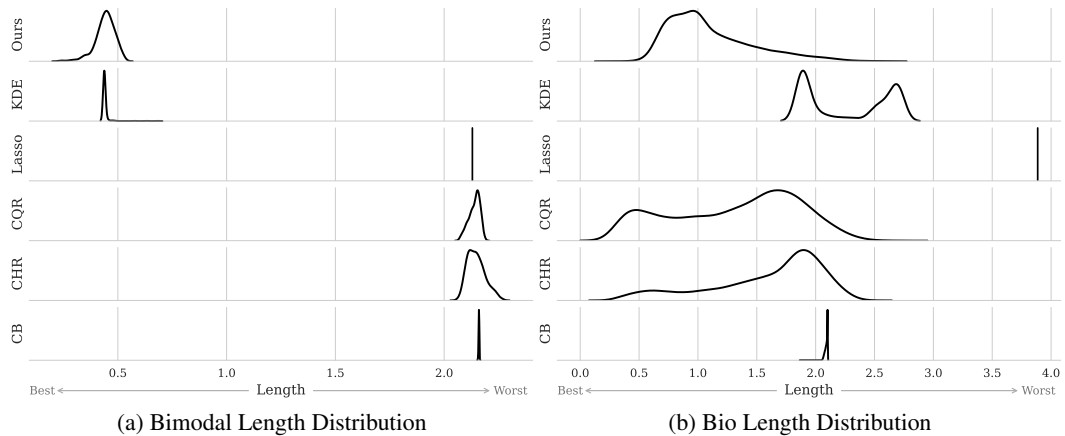

(a) Bimodal Length Distribution

(b) Bio Length Distribution

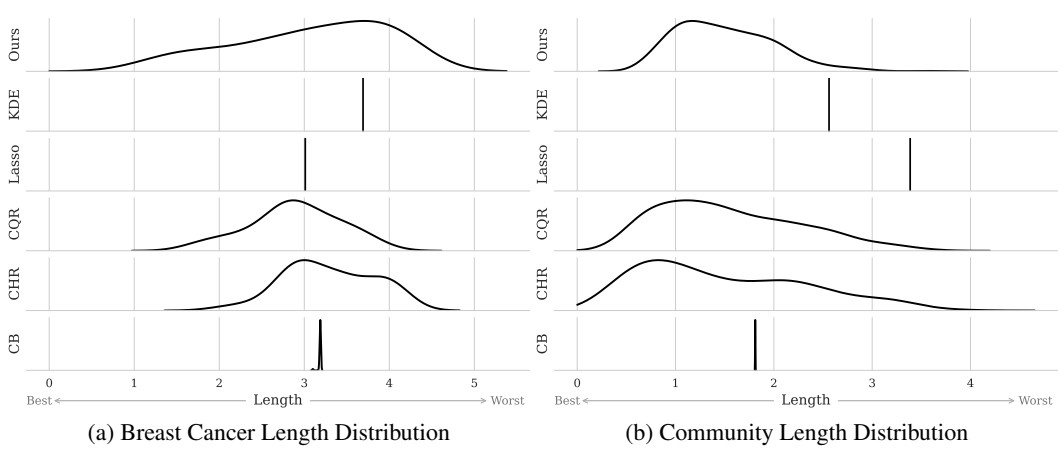

(a) Breast Cancer Length Distribution

(b) Community Length Distribution

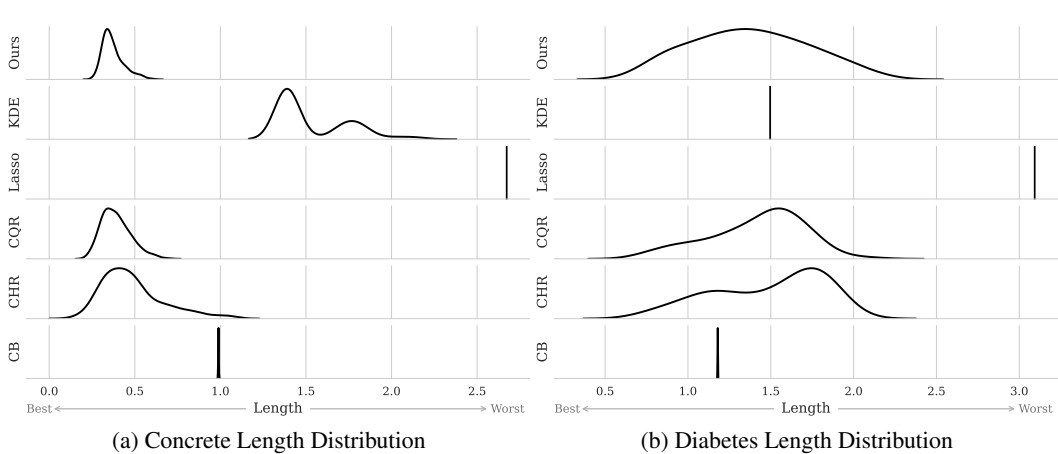

(a) Concrete Length Distribution

(b) Diabetes Length Distribution

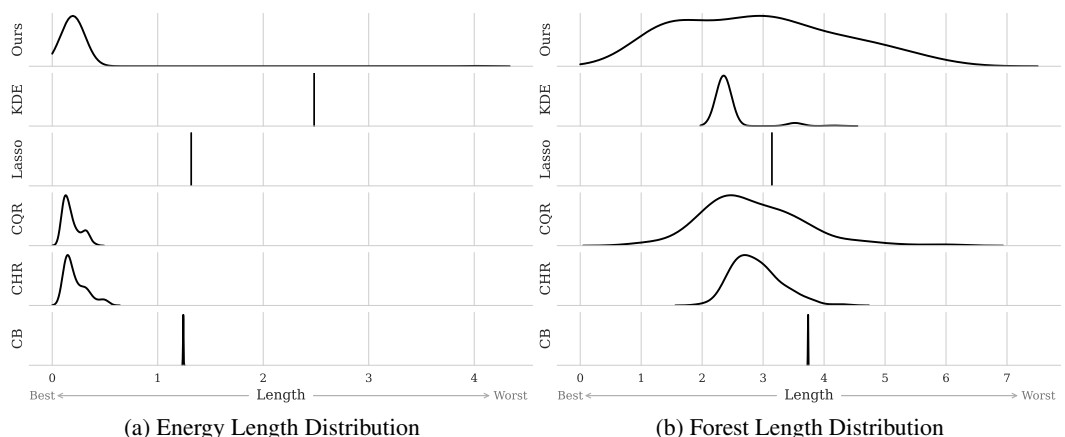

(a) Energy Length Distribution
(b) Forest Length Distribution

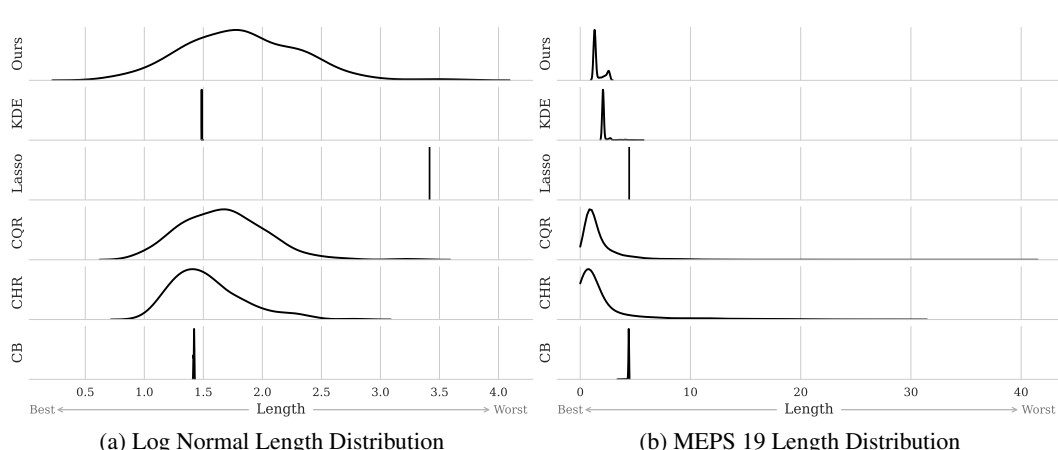

(a) Log Normal Length Distribution
(b) MEPS 19 Length Distribution

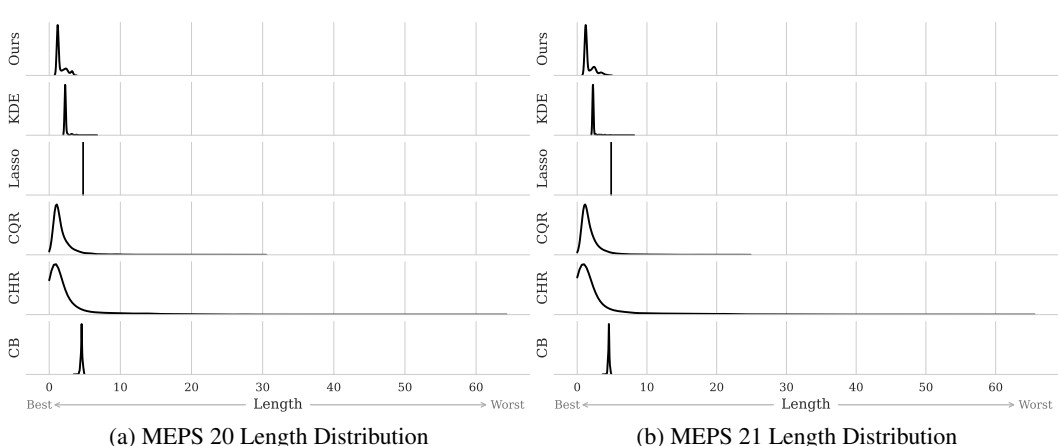

(a) MEPS 20 Length Distribution
(b) MEPS 21 Length Distribution

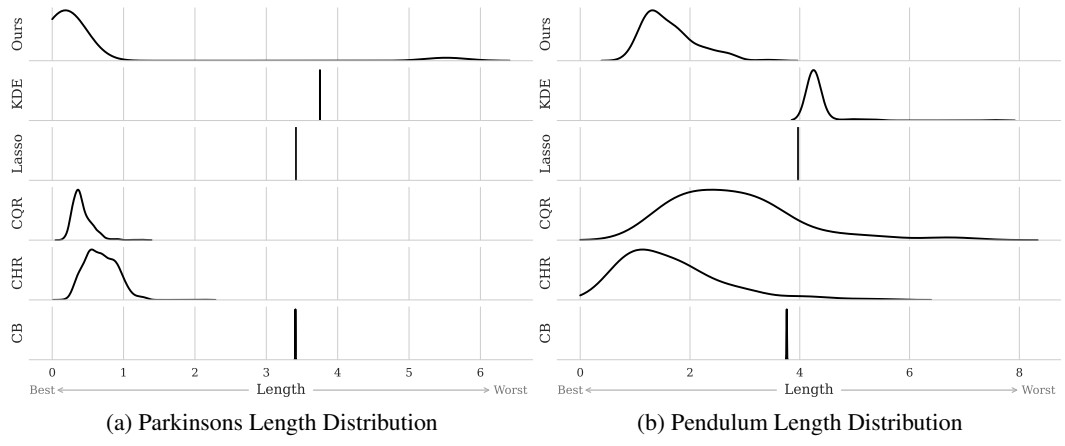

(a) Parkinsons Length Distribution    (b) Pendulum Length Distribution

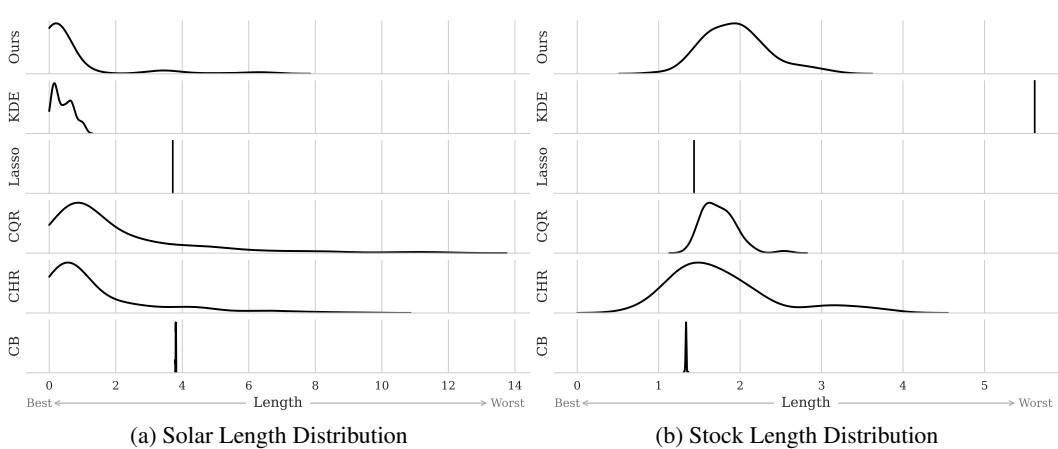

(a) Solar Length Distribution    (b) Stock Length Distribution

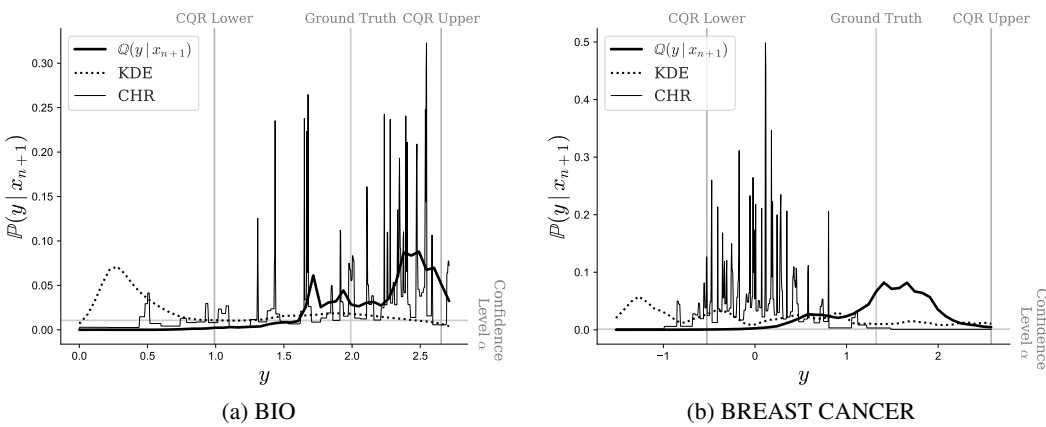

(a) BIO    (b) BREAST CANCER

Figure 7: Plot of outputted density functions for ours, KDE, and CHR on for BIO and BREAST CANCER

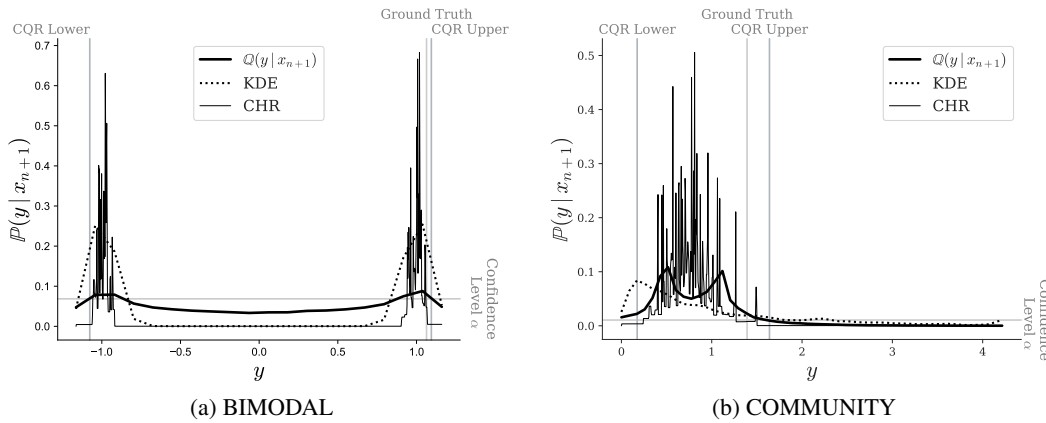

Figure 8: Plot of outputted density functions for ours, KDE, and CHR on for BIMODAL and COM-MUNITY

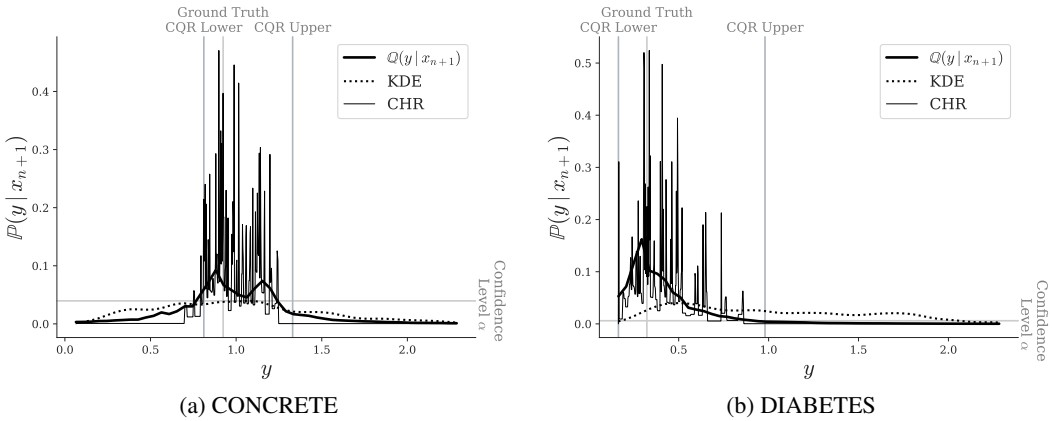

Figure 9: Plot of outputted density functions for ours, KDE, and CHR on for CONCRETE and DIABETES

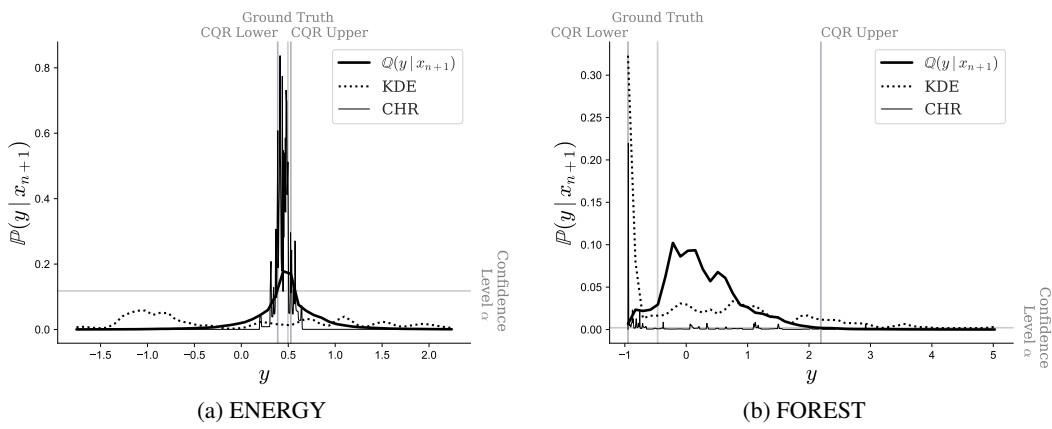

Figure 10: Plot of outputted density functions for ours, KDE, and CHR on for ENERGY and FOREST

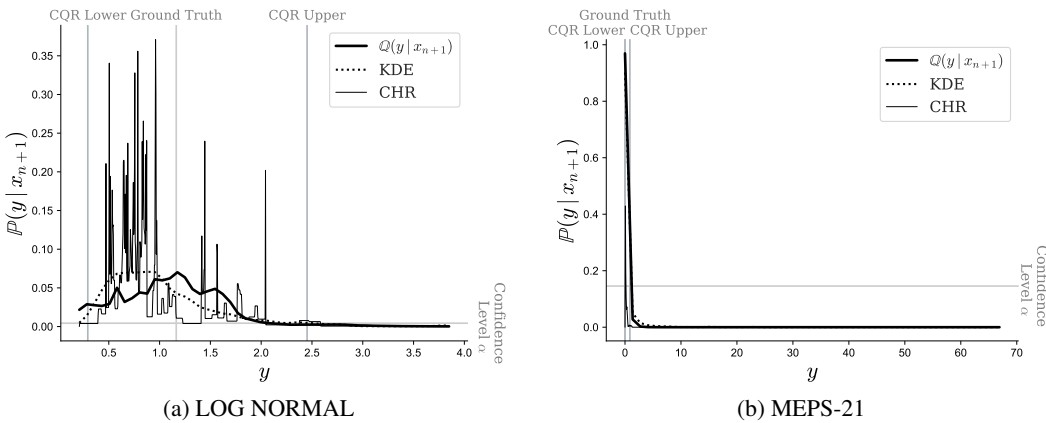

Figure 11: Plot of outputted density functions for ours, KDE, and CHR on for LOG NORMAL and MEPS-19

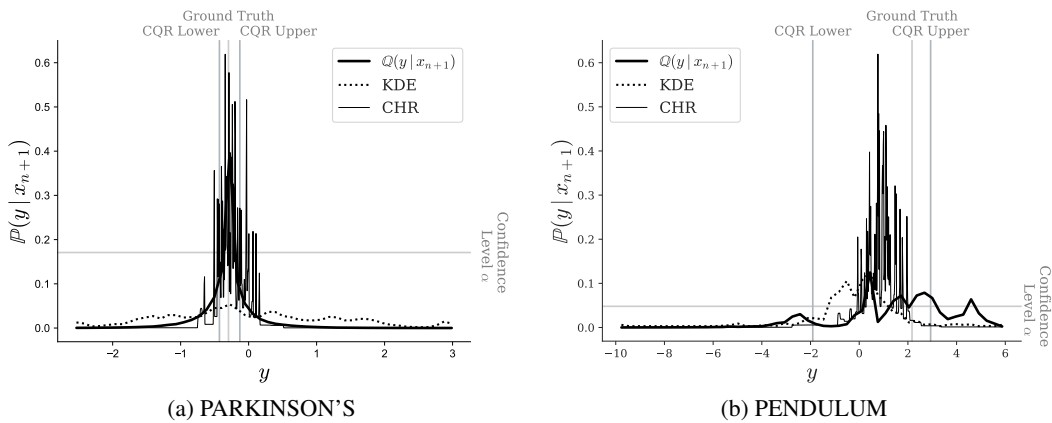

Figure 12: Plot of outputted density functions for ours, KDE, and CHR on for PARKINSON'S and PENDULUM

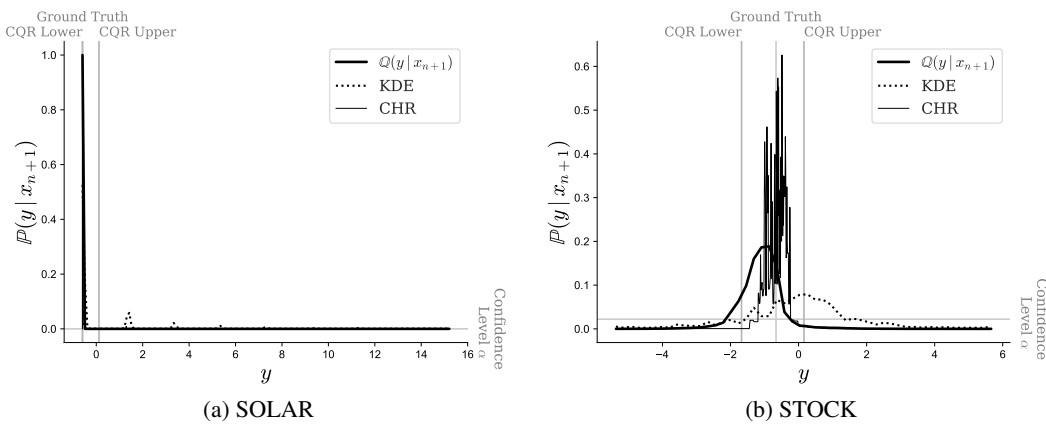

Figure 13: Plot of outputted density functions for ours, KDE, and CHR on for SOLAR and STOCK

