# OpenReview forum: "Conformal Prediction via Regression-as-Classification"
_ICLR.cc/2024/Conference — ICLR 2024 poster_

### Official Review · Reviewer_ZE11 · 2023-10-25

**Soundness:** 3 good
**Presentation:** 3 good
**Contribution:** 2 fair
**Rating:** 3
**Confidence:** 5

**Summary:**

This paper presents a new method to model the conditional density in heteroscedastic regression problems. The idea is to convert metric regression to an ordinal regression problem, in which the conditional density is approximated by a number of fixed bins. This is in essence a histogram approach. Unlike most ordinal regression methods, the authors do not consider an underlying latent continuous variable, but they model every bin via a separate neuron propagating from the embedding layer. A softmax operation is used to guarantee that a valid probability density function is returned, but, unlike classification methods, a loss function that incorporates the order in the bins is used during training.

In a conformal prediction type of experiments, the authors show that this new model results in better prediction sets compared to some baselines for modelling the conditional density, such as quantile regression and kernel density estimation.

**Strengths:**

- The paper is well written and easy to follow
- The paper has some novelty, although I would argue that the core idea is just an application of reference Stewart et al.
- I enjoyed reading this paper, and I am convinced that the method can work relatively well for regression problems with multimodal conditional density distributions
- I liked that some limitations of the method are discussed at the end. However, I do see other limitations as well.

**Weaknesses:**

I do see several important weaknesses.

1. Contributions:
After reading the introduction it was not clear to me what the contributions are. This only became clear to me after reading the experiments. Contributions should be clear from the beginning. To my opinion, the main contribution is proposing a new nonparametric method for modelling the conditional density. However, I don't see any contribution w.r.t. conformal prediction. In the experiments, the authors only consider the standard split conformal prediction, which is arguably not the most useful type of evaluation for heteroscedastic regression problems. I would argue that conditional coverage guarantees need to be analyzed in such cases. This can be established via Mondrian conformal prediction or normalized nonconformity scores, such as normalized absolute residuals. Estimates of the variance could be obtained with any method that models conditional densities, or with methods that directly estimate the variance, such as mean-variance neural networks. So, in that regard, I find the scope and the experimental setup quite limited.

2. Baselines:
The authors consider a few other methods for estimating the conditional density, such as kernel density estimation and quantile regression. However, many other methods exist for estimating the conditional density. Since this is the key contribution of the paper, I would like to see a more thorough experimental and theoretical comparison with such methods. Examples of such methods are mean-variance neural networks, conditional transformation models and methods based on normalizing flows. The proposed method estimates the conditional density via a specific histogram. Methods based on nearest neighbors and regression trees also produce histogram-style conditional densities. Another approach to model multimodal distributions is via Gaussian mixtures. So, there is a lot out there already... Where should we situate the new method is this diverse landscape? I would classify the new method as yet another way of modelling the conditional density. Not better or worse than others, perhaps useful in specific situations, such as multimodal distributions, but more discussion is needed.

3. Limitations of the method:
- By modelling the conditional density as a histogram, one obtains a very non-smooth approximation of the density. So, that's why interpolation is needed to obtain good nonconformity scores. Some of the other methods that are described above don't have this limitation.

- The number of bins looks like a very tricky hyperparameter to tune. The performance probably highly depends on that hyperparameter. What would be a good way to tune this hyperparameter? Another tricky hyperparameter is the level of entropy regularization. I would assume that the results are also very sensitive to that hyperparameter.

- I believe that the proposed method might work reasonably well for multimodal distributions that are not too complex, but I would not recommend this method for unimodal regression problems. For such problems, there are too many degrees of freedom.

4. Unclear aspects of the experiments
- In Table 3 a lot of standard deviations are zero, or close to zero. How is that possible? What is randomized? With randomizations of the testset these numbers are unrealistically small I would say.
- Table 1 versus Table 3: normally one would see that the length of the intervals increases when the coverage increases. So, the two criteria highly depend on each other. Any comparison of two methods should always make the trade-off between those two criteria.

**Questions:**

See above.
In addition, what's novel in this paper compared to reference Stewart et al.?

---

> ### Author Response · Authors · 2023-11-18
> **Rebuttal to Reviewer ZE11 (part 1)**
>
> We thank this reviewer for their thoughtful review.
> We believe that  the reviewer has several confusions regarding some of the concepts regarding conformal prediction methodologies relevant to our work. We hope this response clears up all of the confusions and we will greatly appreciate an increase in score!
>
> > *the main contribution is proposing a new nonparametric method for modelling the conditional density. However, I don't see any contribution w.r.t. conformal prediction*
>
> Most papers on CP  focus on the choice of score function. Depending if one want to consider heteroscedasticity, bimodality, skewness etc. Our contribution single handedly covers these case contrarily to DCP, CQR, CHR.
>
> > *experimental and theoretical comparison with other methods for estimating the conditional density*
>
> The goal of our work is to improve conformal prediction, rather than proposing a better method for conditional density estimation (which is a challenging problem in itself). This is why we use best performing CP methods as baselines. Showing that better conditional density leads to better CP method would be an interesting direction, but would lead to a different paper.
>
> > *Why not add additional baselines such as mean-variance networks?*
>
> These methods of Conditional density estimation have not been used in CP literature before. While they are reasonable ways to estimate conditional density, our method is very simple. There are many ways to do conditional density estimation like you mentioned (see Rothfuss et al.). However, these are not any simpler and are more involved than our technique. Moreover, the existing CP baselines do not compare with these techniques either. We do compare with methods like KDE or Histogram Regression which are both conditional density estimators and the latter uses neural networks. Relative to relevant papers such as CHR, we compare to the same amount or more baselines, so we believe our experimental setup is sufficient.
>
> > *The standard split conformal prediction, which is arguably not the most useful type of evaluation for heteroscedastic regression problems.*
>
> We are afraid that the reviewer is confusing split conformal methods (which is an alternative to the highly expensive Full conformal methods) and the choice of the score function that account for heteroscedasticity. For example, CQR or DCP explicitly account for heteroscedasticity and comes with both Split and Full version, where the former is preferred for computational efficiency.
>
> > *Conditional coverage guarantees need to be analyzed in such cases. This can be established via Mondrian conformal prediction*
>
> There are several types of conditional guarantees Mondrian CP are designed to be conditional on the class of the input object. In this case, the p-values function itself is modified by including a specific taxonomy (e.g. provided by a classifier or clustering algorithm). These are available in both the Split CP version and the Full CP version (or even other cross-conformal alternatives). Being split or not is about computational efficiency. Heteroscedasticity or other properties are handled by the choice of the conformity score function.
>
> We would like to add that CQR provides a benchmark against variance-based normalization and achieved superior performance.
>
> By conformalizing our density estimation leveraging a (Mondrian or Venn) taxonomy, our method trivially accounts for class conditionality since this is independent of the choice of score function (it is just about the choice of p-value function). For a review of these concepts, please refer to Chapter 4.6.1 of the book 'Algorithmic Learning in a Random World', second edition, by Vovk, Gammerman, and Shafer.
>
> Moreover, most of the baselines in regression CP focus on split conformal prediction. It is a standard technique in the CP literature. We believe it is unfair to reject a paper for using a standard evaluation setup.
> ## References
>
> Jonas Rothfuss, Fabio Ferreira, Simon Walther, and Maxim Ulrich. Conditional density estimation
> with neural networks: Best practices and benchmarks. arXiv preprint arXiv:1903.00954, 2019.

---

> ### Author Response · Authors · 2023-11-18
> **Rebuttal to Reviewer ZE11 (part 2)**
>
> > *We learn nonsmooth distribution but other papers don’t.*
>
> If you look at Figure 4 in the appendix, we learn pretty smooth distributions out of the box. Interpolation is *not* used for this. It is only used to make a discrete density function into a continuous one. Our interpolation is a simple linear interpolation, which does no extra smoothing on top. In fact, the smoothness is a strength of our method. If you look at CHR, they learn very spiky distributions.
>
> > *You are sensitive to hyperparameter tuning.*
>
> No tuning was done at all. We hold $K=50$ bins and $\tau=0.2$ across all experiments which span 16 different datasets with widely varying distribution shapes. If tuning was done, we would expect even better results. We will add ablations in the final version of the paper to investigate how our results are dependent on these hyperparameters.
>
> > *This won’t work on unimodal distributions.*
>
> This is not correct. Many of the data such as MEPS are unimodal and we get the best results out of all methods.
>
> >  *There are unclear aspects of the experiments*
>
> We repeat the experiments over different seeds and report the average mean over all seeds. The value you are referring to in Table 3 is the standard error (we have clarified this in the uploaded version). Since we use a large proportion for the test set, this suggests that our method performs well across numerous seeds, and small standard errors are reasonable.
>
> > *There is a tradeoff between coverage and length*
>
> We have fixed all experiments at coverage level of $90\%$. We have specified this in newly uploaded versions. This is standard in the CP literature.
>
> > *How does this differ from Stewart et al.?*
>
> Stewart et. al focuses on learning conditional expectation while we focus on conditional density. Using their loss function naively performs worse than our smoothened version of the loss function according to the ablations. We will clarify better in the final version that Stewart et al’s loss is included in the ablations and did not work. Our loss function is tailored to our specific problem, for example with smoothing.

---

> ### Author Response · Authors · 2023-11-23
> **End of discussions period**
>
> Dear Reviewer ZE11,
>
> The discussion period is about to conclude, and despite your did not engage with us yet, we are happy that you enjoyed reading our paper and find it convincing for regression problems with multimodal distributions.
>
> Unfortunately, you have the lowest score among the reviewers, and the highest confidence.
>
> We have diligently replied to all your concerns and several confusions, in detail in our rebuttal, and we would be delighted to provide any further clarifications. We believe that not all the limitations you have expressed are well-founded, and this certainly explains your rating. If our response is not satisfactory, please let us know asap, and we will be happy to add further details in the short time available.
>
>
> Best regards,
>
> Authors

---

### Official Review · Reviewer_iHe5 · 2023-10-29

**Soundness:** 3 good
**Presentation:** 3 good
**Contribution:** 3 good
**Rating:** 8
**Confidence:** 4

**Summary:**

The paper proposes to convert 1D regression problems into K-class classification problems by partitioning the label space into K bins.  This
allows leveraging known techniques for classification CP. Like Lei et al. 2004, the authors use the conditional density produced by the classification model as a conformity score. On top of it, a regularization model is trained to enforce smoothness when the discrete distribution is converted back to a continuous one.

**Strengths:**

I like the idea of learning the conformity score is interesting.  Works like [1] also use density estimation to adjust the conformity score. But the authors' approach is intrinsically different.  The proposed objective function seems a good alternative to optimizing the efficiency of the prediction intervals.

**Weaknesses:**

The role of the discretization step may be explained better. The authors could emphasize the difference with other techniques for estimating conditional densities or explain why the proposed approach does not suffer from the usual instability of standard estimators.

**Questions:**

- Is the smoothness-enforcing penalty new?
- How were K and Tau chosen? Is part of the data used for training the conditional distribution?
- Why should the entropy regularization be expected to be good at learning bi-modal distributions?
- The size of the prediction intervals is a good measure of efficiency. Another one is the correlation between the test errors and the corresponding intervals. It would be interesting to see a scatter plot of the absolute residual versus the prediction intervals in a couple of data sets where R2CCP is or is not the best algorithm.
- Is the proposed approach equivalent to training a parameterized conformity function (see for example in [2] or [3])?

[1] Guan, Localized conformal prediction: A generalized inference framework for conformal prediction (2023)
[2] Einbinder et al., Training Uncertainty-Aware Classifiers with
Conformalized Deep Learning (2022)
[3] Colombo, On training locally adaptive CP (2023)

---

> ### Author Response · Authors · 2023-11-17
> **Answer Reviewer iHe5**
>
> We thank this reviewer for their thoughtful review. We clarify your weaknesses and questions here. If you find this sufficient, we would greatly appreciate an increase in score!
>
> ## Weaknesses
>
> > *What is the role of discretization in this work?*
>
> Several works in the literature, including Stewart et al. mention that the classification problem tends to lead to stabler training than regression.  Stewart et al. empirically demonstrates that R2C estimates the conditional expectation better than regression alone. The reason that this is the case is an open theoretical problem. However, intuitively, the learning problem under R2C is an unconstrained learning problem with only $K$ outputs whereas methods like CHR have to have 1000 output nodes and average among them to achieve smoothness.
>
> ## Questions
>
> > *Is the smoothness-enforcing penalty new?*
>
> Smoothing-enforcing penalty has existed in other contexts such as Variational Learning. However, in this context of learning conditional densities using Neural Networks, it is new to the best of our knowledge.
>
> > *How were K and Tau chosen? Is part of the data used for training the conditional distribution?*
>
> $K $, $p$, and $\tau$ were fixed at $50$, $.5$, and $.2$ for all experiments. We didn't change that or tune that as that would be unfair to other methods. Tuning these would likely improve the results of our method.
>
> > *Why should the entropy regularization be expected to be good at learning bi-modal distributions?*
>
> It’s true. It is possible that reducing entropy can sometimes help, but as a general tool, entropy regularization helps overall. Empirically, our structure still works well for bimodal distributions.
>
> > *Is part of the data used for training the conditional distribution?*
>
> We are not sure we understand the question. As usual, the dataset was split in two (training and calibration), and indeed, the training set was used to learn the conditional distribution.
>
> > *What is the correlation between the test errors and the corresponding intervals?*
>
> Thanks for your suggestion. We will try to illustrate this plot for a couple of benchmarks for the camera ready version.
>
> > *Is the proposed approach equivalent to training a parameterized conformity function (see for example in [2] or [3])?*
>
> We are very different from the Einbinder paper since their loss function tries to ensure its scores form a smooth CDF during training and is nondifferentiable. We do no such thing and use much simpler differentiable methods. The second Colombo paper proposes how to transform conformity scores to make it more suitable for APS.
>
> Thanks for the additional reference, we will add [1] in our related work section.
>
> ## References
>
> Lawrence Stewart, Francis Bach, Quentin Berthet, and Jean-Philippe Vert. Regression as classifi-
> cation: Influence of task formulation on neural network features. In International Conference on
> Artificial Intelligence and Statistics (AISTATS), 2023.

---

### Official Review · Reviewer_HZzS · 2023-10-29

**Soundness:** 3 good
**Presentation:** 3 good
**Contribution:** 3 good
**Rating:** 8
**Confidence:** 4

**Summary:**

The paper proposes to see a regression problem as a classification one, and then to apply ideas issued from conformal prediction in order to derive predictions regions that may not be compact sets (here, union of intervals).

**Strengths:**

+: an interesting view of the problem, especially as this allows the conformal predictions to be easily something else than intervals, something that may indeed be of important interest for regression. No other conformal regression methods that I know of achieve this kind of things, with maybe the exception of "Conformal prediction in manifold learning" (not cited by the authors, but the paper is only weakly related to the present work), since intervals on the manifold may well turn out to be non-compact in the original space.

+: a quite well written paper, and a method simple enough to be applicable in a wide range of settings.

+: experiments that are convincing enough to show the interest of the method.

**Weaknesses:**

-: the way authors frame the regression problem as a classification is very, very close to the standard ordinal regression problem, and I really missed some positioning with respect to this literature. There is a huge literature on ordinal regression (maybe look at a general paper, e.g., "Tutz, G. (2022). Ordinal regression: A review and a taxonomy of models. Wiley Interdisciplinary Reviews: Computational Statistics, 14(2), e1545."), but also a couple papers using ordinal conformal classification (see questions).

**Questions:**

- Could you position the current paper with respect to papers dealing with ordinal conformal regression? Two I know of (there could be others, but not much more) are "Xu, Y., Guo, W., & Wei, Z. (2023, July). Conformal Risk Control for Ordinal Classification. In Uncertainty in Artificial Intelligence (pp. 2346-2355). PMLR." and "Lu, C., Angelopoulos, A. N., & Pomerantz, S. (2022, September). Improving trustworthiness of ai disease severity rating in medical imaging with ordinal conformal prediction sets. In International Conference on Medical Image Computing and Computer-Assisted Intervention (pp. 545-554). Cham: Springer Nature Switzerland.".

- It seems to me that the conformal scores used in Algorithm 1 mostly rely on modal values. In ordinal regression, it is much more common to use the average rank (under $L_2$ loss) or the median (under $L_1$ loss) as predictions. Could you comment on such options?

- It would be nice to have an idea of how often the output predictions regions are not interval, to have an idea of how often we depart from classical methods. Would it be possible ot have an idea (in the appendix or main paper).

- Could you display full coverage curve (say, from 90% to 99%) rather than table for a fixed value? (side remark: there is a double ?? in appendix B).

- Bonus question for myself: do you have an idea of how the present method could be adapted to multi-variate output settings?

**Details Of Ethics Concerns:**

not needed

---

> ### Author Response · Authors · 2023-11-17
> **Answer to Reviewer HZzS**
>
> We thank the reviewer for their thoughtful review. We clarify the weaknesses and questions here. If you find this sufficient, we would greatly appreciate an increase in score!
>
> ## Weaknesses
>
> > *Can you connect more to ordinal regression?*
>
> Thanks for the suggestion. We completely agree with you that there are strong connections with ordinal regression. We noted on this connection with citations to Weigend & Srivastava (1995) and Diaz & Marathe (2019). More recently, in computer vision, also some investigation is done to understand the effectiveness of such methods (https://arxiv.org/pdf/2301.08915.pdf). We have also added several relevant citations to the ordinal regression literature in the related works in the newly uploaded paper.
>
> ## Questions
>
> > *How does this paper compare to conformal ordinal classification papers?*
>
> We have added these two new works to the related works in the newly uploaded document. “Conformal Risk Control for Ordinal Classification” talks about different risk types (analogous to coverage) for ordinal classification while we discuss different score functions when coverage is your loss function. “Improving Trustworthiness of AI Disease Severity Rating in Medical Imaging with Ordinal Conformal Prediction Sets” talks about how you can adapt APS style conformal sets to work with ordinal structure, which was later used in the CHR paper.
>
> > *Why do we focus on modal values?*
>
> A key contribution of our paper is its versatility across different distributions via building a probability density function. Consider a bimodal label distribution; the median and mean would be very unlikely answers since they lie in the middle of two valleys. To account for such distributions, we found modal values to be a better fit.
>
> > *How many predictions are not intervals?*
>
> We can do this in the final version of the paper.
>
> >  *Can you do a full coverage curve?*
>
> We have fixed the broken reference in the appendix in the newly uploaded version. Moreover, we can work on the full coverage curve for the camera-ready version.
>
> > *Can this be extended to multi-variate output?*
>
> Our method can be adapted to multivariate outputs, and this is an exciting direction for future work. There is a direct naive approach, which scales like $K^d$, where $K$ is the number of bins and $d$ the dimension of the output. Developing more efficient ways which do not scale exponentially with the dimension of the output (but perhaps like $O(K*d)$) is not straightforward, but can be an interesting direction for future work. Thanks for the question!
>
> Raul Diaz and Amit Marathe. Soft labels for ordinal regression. In IEEE Conference on Computer
> Vision and Pattern Recognition (CVPR), 2019.
>
> Andreas S Weigend and Ashok N Srivastava. Predicting conditional probability distributions: A
> connectionist approach. International Journal of Neural Systems, 6(02):109–118, 1995.

---

> > ### Comment · Reviewer_HZzS · 2023-11-23
> > **Thanks for the feedback + sorry for the lack of time to interact**
> >
> > Dear authors,
> >
> > Thanks for the detailed feedback and answers, as well as for the minor modifications in the main body of the paper. Unfortunately I am currently involved in lab evaluation committees which take up to ten hours of work daily, and does not have the time to really engage in time-constrained, online discussions (such things are better suited to journal publications that are less time-pressured). I apologize for this, but will take a closer look at the answers and changes to make a final recommendation.
> >
> > However, as my questions were mostly clarification questions or small updates, and as I was already quite positive about the paper, I would not consider that the answers and change have signifincalty increased the quality of the paper. Also, while I appreciate the authors detailed answer and politeness, I would like to note that their request to reviewers to change scores may actually be counter-productive.
> >
> > Best regards

---

### Official Review · Reviewer_RG4V · 2023-11-01

**Soundness:** 3 good
**Presentation:** 3 good
**Contribution:** 2 fair
**Rating:** 6
**Confidence:** 4

**Summary:**

The paper proposes a new way to do conformal prediction for regression based on conformalizing a regression-to-classification method. It is claimed that the new method offers higher flexibility compared to using a traditional regressor, making it more suitable for heteroscedastic or multimodal data.

**Strengths:**

- The paper tackles a traditional conformal problem, but still seems to offer improvements to the state-of-the-art.
- The method proposed is interesting; I had not previously seen this regression-as-classification setup in literature and appreciate the authors providing a number of references to the broader framework.
- The paper is easy to follow, although there are some issues which I will point out in weaknesses. I also noticed the choice to not state and/or highlight the conformal coverage guarantee, which is often oversold nowadays. (I still think you should state the guarantee at least in the Appendix so that readers unfamiliar with conformal prediction are aware of it.)

**Weaknesses:**

## Writing
Some parts of the paper were difficult to follow:
- The proposed method was not easy to understand before seeing Algorithm 1. I felt Section 3 had too many comments on "motivation" that do not focus on the actual method and are a bit counterproductive to a quick read. E.g., Sec 3.1, "We  aim to compute... classification context" does not add much. In the end, you have a traditional neural network with softmax output, which everyone understands. Similarly the comment "This approach is both straightforward and efficient ... information or structure." does not add anything substantial. Sec 3.2 "It would be desirable to be able to use similar methods for both classification and regression conformal prediction". This is a somewhat subjective (and distracting) claim that I believe is best avoided. The paragraph starting with "These values yˆ ∈ Yˆ f..." is unnecessarily verbose. Sec 3.3 introduces a loss function without giving an example of what it could be.
- Table 1 is very hard to understand since (i) the methods are not connected to the acronyms, (ii) it is not clear what is inside the brackets, (iii) length and coverage is not defined in the paper.

## Technical concerns
- Most results are not reported with error bars making them less reliable. For instance, in Table 1, CHR does worse than CQR but it should not? (both methods are proposed by similar authors and CHR comes later).
- Have you considered a baseline where you learn a regressor directly using the same neural network architecture, treat that regressor as \bar{q}, and apply the same conformalization? It is similar to the "optimal prediction sets" in Appendix F here: https://arxiv.org/pdf/1910.10562.pdf .

I am not familiar with the regression-as-classification (R2C) literature. The main concern I have is that it is unclear which aspects of the R2C method are novel and which ones directly derive from previous work. Have you made significant changes to previous R2C methods to adapt them to the conformal problem? The key relevant technique seems to be smoothing; has these been proposed before?

**Questions:**

- Could you elaborate on what linear interpolation is used to go from q_\theta to \bar{q}_\theta? It is not clear to me at all.
- You have pointed out a number of papers which advocate for regression-as-classification. However, it is still not convincing as to why we should first destroy the ranking structure of reals, and then reinstate it using the entropic regularization. Why not just do usual regression?
- Which aspects of the R2C method are novel and which ones directly derive from previous work? Have you made significant changes to previous R2C methods to adapt them to the conformal problem? The key relevant technique seems to be smoothing; has these been proposed before?

---

> ### Author Response · Authors · 2023-11-17
> **Answer to Reviewer RG4V**
>
> We thank this reviewer for their thoughtful review. We clarify your weaknesses and questions here. If you find this sufficient, we would greatly appreciate an increase in score.
>
> ## Weaknesses
>
> > *The proposed method was not easy to understand before seeing Algorithm 1*
>
> Although we understand your concern, we do not want to assume that everyone is familiar with conformal prediction and its connection to our estimated model. Thus, we first expand on what motivates our proposal and connect it to the existing literature before diving into more technical details. We will consider some of your suggestions to clarify the final version of the paper.
>
> > *Table 1 is very hard to understand*
>
> Thanks, we agree with your points. We have fixed these in the newly uploaded version.  We have specified that the brackets include the “Standard Error” over the experiments. We also provide definitions for length and coverage. We believed the graphs and terms were standard enough in the literature. To avoid confusion, we will make sure to be more specific in the final submission.
>
> > *Most results are not reported with error bars making them less reliable. For instance, in Table 1, CHR does worse than CQR but it should not?*
>
> We run all of our experiments by averaging over the test set and over 5 different seeds. With every different seed, we retrain the model. The standard error of the results over the different seeds is shown in the brackets. Given that we are running experiments across significantly more datasets than the ones used in the CHR and CQR papers, it is reasonable that upon seeing new datasets, CHR does worse than CQR in some cases, and in some cases, CHR outperforms CQR. For both baselines, we use code written by the authors themselves and do not edit them at all. **We report the numbers given by their open-sourced code explicitly.**
>
>
> >  *Have you considered a baseline where you learn a regressor directly using the same neural network architecture, treat that regressor as $\bar{q}$, and apply the same conformalization?*
>
> From what we are aware of, learning conditional density for regression problem using neural network require more involved methods with specific prior such as Gaussian mixture parametrization or Bayesian network (see Rothfuss et al.). Here, we are interested in computing a confidence set instead of the harder problem of getting accurate density estimation. As such, we propose to bypass those difficulties by converting the problem into a multiclassification problem where the parametrization is simpler. As you stated above, we rely on  “a traditional neural network with softmax output, which everyone understands”. Also, we note that CHR uses a histogram powered by a neural network to estimate the conditional density in a regressor setting, as you suggest. On top of that, we use 6 baseline effective conformal methods in our benchmark. Of course, we can always add more, but we believe that our numerical experiments are revealing enough.
>
> > *Which aspects of the R2C method are novel?*
>
> As is presented, this method of R2C for conformal prediction has not been done before. Smoothing in different contexts has been proposed, such as in ordinal regression, but not in R2C literature. However, Stewart et al. is the standard R2C loss function, and using that directly does not work (see our ablation studies). We will make it clearer that the loss function from Stewart et al. is the one studied in the ablations. They focus on conditional expectation, whereas our loss focuses on conditional density. Our smoothened loss function works better according to the ablations. This use of R2C to learn conditional density alongside smoothing is novel to this paper.
>
> ## Questions
>
> > *What is the exact formulation of linear interpolation*
>
> We have added this in the newly uploaded version. We clarify it as below (note here the $\hat{y_k}$ and $\hat{y_{k+1}}$ are chosen such that  $\hat{y_k} < y < \hat{y_{k+1}}$:
>
>
> $\bar{q}(y \mid x) = $ $\frac{(\hat{y_{k + 1}} - y) q(\hat{y_k} \mid x)}{\hat{y_{k + 1}} - \hat{y_k}}  $ $+ \frac{(y - \hat{y_{k}}) q(\hat{y_{k+1}} \mid x)}{\hat{y_{k + 1}} - \hat{y_k}} $
>
> > *Why use R2C versus standard regression?*
>
> Several techniques exist to use regression techniques to learn conditional densities, as shown in our baselines. However, these methods can suffer from poor stability during training. For example, see Figure 4, and you will see that CHR’s technique using regressors is not smooth. Our loss function results in smoother and more stable densities. In fact, the Stewart et al. paper demonstrates empirically where R2C outperforms regression techniques for estimating the conditional expectation. Why this is the case theoretically is still an open research problem, as far as we know.

---

> ### Author Response · Authors · 2023-11-18
> **References**
>
> Lawrence Stewart, Francis Bach, Quentin Berthet, and Jean-Philippe Vert. Regression as classifi-
> cation: Influence of task formulation on neural network features. In International Conference on
> Artificial Intelligence and Statistics (AISTATS), 2023
>
> Jonas Rothfuss, Fabio Ferreira, Simon Walther, and Maxim Ulrich. Conditional density estimation
> with neural networks: Best practices and benchmarks. arXiv preprint arXiv:1903.00954, 2019.

---

> ### Author Response · Authors · 2023-11-23
> **End of discussions**
>
> Dear Reviewer RG4V,
>
> We appreciate your review and would be grateful if you could let us know if our response resolved the issues you raised.
> We regret that the review period is ending and we have not had the opportunity to discuss your questions further with you. Consequently, we are concerned that your rating is still below the acceptance threshold.
>
> If our response was satisfactory, please leave a comment and update your score accordingly. We would be pleased to provide further explanations if desired.
>
> Best,
>
> Authors

---

### Author Response · Authors · 2023-11-22
**Discussions**

Dear Reviewers,

Thank you again for your detailed feedback.

We believe we have addressed the main concerns and invite you to engage with us with any remaining questions or unclear parts about our paper. Please note that we cannot update the paper after Wednesday, the 22nd of November, 2023.

Best regards,

Authors

---

### Comment · Area_Chair_mHC5 · 2023-12-03
**Two followup questions**

Dear Authors':

Can you please respond to the following two questions?

1. The paper didn't compare to recent CP methods based on the principle of localization, which are primarily aimed at reducing the prediction interval length.

- Locally Valid and Discriminative Prediction Intervals for Deep Learning Models, NeurIPS-2021.
- Improving Uncertainty Quantification of Deep Classifiers via Neighborhood Conformal Prediction: Novel Algorithm and Theoretical Analysis, AAAI-2023.

Please provide qualitative and/or quantitative analysis to answer this question.

2. The number of bins seem like an important hyper-parameter and one needs a way to select it for a given regression task. What is the methodology for tuning it?

Please provide some representative quantitative analysis to demonstrate coverage and prediction interval trade-off as a function of the number of bins using the methodology listed to answer this question.

Thanks,
Your AC

---

### Meta-Review · Area_Chair_mHC5 · 2023-12-09

**Metareview:**

This paper considers the problem of improving conformal prediction (CP) for regression tasks to reduce the prediction interval length for heteroscedastic data. The overall approach is based on a reduction from regression-to-classification that allows one to leverage CP methods for classification.

All the reviewers' are positive about the overall approach, but there is one point that needs to be addressed and improved:
- The paper should do a better job of discussing the challenge of estimating conditional density as a function of the increasing number of bins. How does the proposed approach address this challenge? For what class of problems, the proposed method will work and for what problems it will suffer from this challenge.
- To match this qualitative discussion with comprehensive quantitative results where the method is stress-tested on all benchmarks by varying the number of bins from small to very large values. Discuss these results in the context of qualitative discussion.

The paper is borderline and I'm leaning towards acceptance. If the paper is accepted, I strongly encourage the authors to thoroughly address the above two comments.

**Justification For Why Not Higher Score:**

There is one important weakness for this paper in its current form.

**Justification For Why Not Lower Score:**

I'm giving benefit of doubt to the authors' about the outstanding concern.

---

### Decision · Program_Chairs · 2024-01-16

Accept (poster)